# Freshwater monitoring by nanopore sequencing

Lara Urban[1†]*, Andre Holzer[2†]*, J Jotautas Baronas[3], Michael B Hall[1], Philipp Braeuninger-Weimer[4], Michael J Scherm[5], Daniel J Kunz[6,7], Surangi N Perera[8], Daniel E Martin-Herranz[1], Edward T Tipper[3], Susannah J Salter[9], Maximilian R Stammnitz[9]*

[1]European Bioinformatics Institute, Wellcome Genome Campus, Hinxton, United Kingdom; [2]Department of Plant Sciences, University of Cambridge, Cambridge, United Kingdom; [3]Department of Earth Sciences, University of Cambridge, Cambridge, United Kingdom; [4]Department of Engineering, University of Cambridge, Cambridge, United Kingdom; [5]Department of Biochemistry, University of Cambridge, Cambridge, United Kingdom; [6]Wellcome Sanger Institute, Wellcome Trust Genome Campus, Hinxton, United Kingdom; [7]Department of Physics, University of Cambridge, Cambridge, United Kingdom; [8]Department of Physiology, Development & Neuroscience, University of Cambridge, Cambridge, United Kingdom; [9]Department of Veterinary Medicine, University of Cambridge, Cambridge, United Kingdom

*For correspondence:
lara.h.urban@gmail.com (LU);
andre.holzer.biotech@gmail.com (AH);
maxrupsta@gmail.com (MRS)

†These authors contributed equally to this work

Competing interests: The authors declare that no competing interests exist.

**Abstract** While traditional microbiological freshwater tests focus on the detection of specific bacterial indicator species, including pathogens, direct tracing of all aquatic DNA through metagenomics poses a profound alternative. Yet, in situ metagenomic water surveys face substantial challenges in cost and logistics. Here, we present a simple, fast, cost-effective and remotely accessible freshwater diagnostics workflow centred around the portable nanopore sequencing technology. Using defined compositions and spatiotemporal microbiota from surface water of an example river in Cambridge (UK), we provide optimised experimental and bioinformatics guidelines, including a benchmark with twelve taxonomic classification tools for nanopore sequences. We find that nanopore metagenomics can depict the hydrological core microbiome and fine temporal gradients in line with complementary physicochemical measurements. In a public health context, these data feature relevant sewage signals and pathogen maps at species level resolution. We anticipate that this framework will gather momentum for new environmental monitoring initiatives using portable devices.

## Introduction

The global assurance of safe drinking water and basic sanitation has been recognised as a United Nations Millennium Development Goal (*Bartram et al., 2005*), particularly in light of the pressures of rising urbanisation, agricultural intensification, and climate change (*Haddeland et al., 2014*; *Schewe et al., 2014*). Waterborne diseases represent a particular global threat, with zoonotic diseases such as typhoid fever, cholera, or leptospirosis resulting in hundreds of thousands of deaths each year (*Prüss et al., 2002*; *Prüss-Ustün et al., 2019*).

To control for risks of infection by waterborne diseases, microbial assessments can be conducted. While traditional microbial tests focus on the isolation of specific bacterial indicator organisms through selective media outgrowth in a diagnostic laboratory, this cultivation process is all too often time consuming, infrastructure-dependent and lacks behind in automatisation (*Salazar and*

**eLife digest** Many water-dwelling bacteria can cause severe diseases such as cholera, typhoid or leptospirosis. One way to prevent outbreaks is to test water sources to find out which species of microbes they contain, and at which levels.

Traditionally, this involves taking a water sample, followed by growing a few species of 'indicator bacteria' that help to estimate whether the water is safe. An alternative technique, called metagenomics, has been available since the mid-2000s. It consists in reviewing (or 'sequencing') the genetic information of most of the bacteria present in the water, which allows scientists to spot harmful species. Both methods, however, require well-equipped laboratories with highly trained staff, making them challenging to use in remote areas.

The MinION is a pocket-sized device that – when paired with a laptop or mobile phone – can sequence genetic information 'on the go'. It has already been harnessed during Ebola, Zika or SARS-CoV-2 epidemics to track the genetic information of viruses in patients and environmental samples. However, it is still difficult to use the MinION and other sequencers to monitor bacteria in water sources, partly because the genetic information of the microbes is highly fragmented during DNA extraction.

To address this challenge, Urban, Holzer et al. set out to optimise hardware and software protocols so the MinION could be used to detect bacterial species present in rivers. The tests focussed on the River Cam in Cambridge, UK, a waterway which faces regular public health problems: local rowers and swimmers often contract waterborne infections, sometimes leading to river closures.

For six months, Urban, Holzer et al. used the MinION to map out the bacteria present across nine river sites, assessing the diversity of species and the presence of disease-causing microbes in the water. In particular, the results showed that optimising the protocols made it possible to tell the difference between closely related species – an important feature since harmful and inoffensive bacteria can sometimes be genetically close. The data also revealed that the levels of harmful bacteria were highest downstream of urban river sections, near a water treatment plant and river barge moorings. Together, these findings demonstrate that optimising MinION protocols can turn this device into a useful tool to easily monitor water quality.

Around the world, climate change, rising urbanisation and the intensification of agriculture all threaten water quality. In fact, access to clean water is one of the United Nations sustainable development goals for 2030. Using the guidelines developed by Urban, Holzer et al., communities could harness the MinION to monitor water quality in remote areas, offering a cost-effective, portable DNA analysis tool to protect populations against deadly diseases.

*Sunagawa, 2017*; *Tringe and Rubin, 2005*). Environmental metagenomics, the direct tracing of DNA from environmental samples, constitutes a less organism-tailored, data-driven monitoring alternative. Such approaches have been demonstrated to provide robust measurements of relative taxonomic species composition as well as functional diversity in a variety of environmental contexts (*Almeida et al., 2019*; *Bahram et al., 2018*; *Tara Oceans coordinators et al., 2015*), and overcome enrichment and resolution biases common to culturing (*Salazar and Sunagawa, 2017*; *Tringe and Rubin, 2005*). However, they usually depend on expensive stationary equipment, specialised operational training and substantial time lags between fieldwork, sample preparation, raw data generation and access. Combined, there is an increasing demand for freshwater monitoring frameworks that unite the advantages of metagenomic workflows with high cost effectiveness, fast technology deployability, and data transparency (*Gardy and Loman, 2018*).

In recent years, these challenges have been revisited with the prospect of mobile DNA analysis. The main driver of this is the 'portable' MinION device from Oxford Nanopore Technologies (ONT), which enables real-time DNA sequencing using nanopores (*Jain et al., 2016*). Nanopore read lengths can be comparably long, currently up to ~$2*10^6$ bases (*Payne et al., 2019*), which is enabled by continuous electrical sensing of sequential nucleotides along single DNA strands. In connection with a laptop for the translation of raw voltage signal into nucleotides, nanopore sequencing can be used to rapidly monitor long DNA sequences in remote locations. Although there are still common

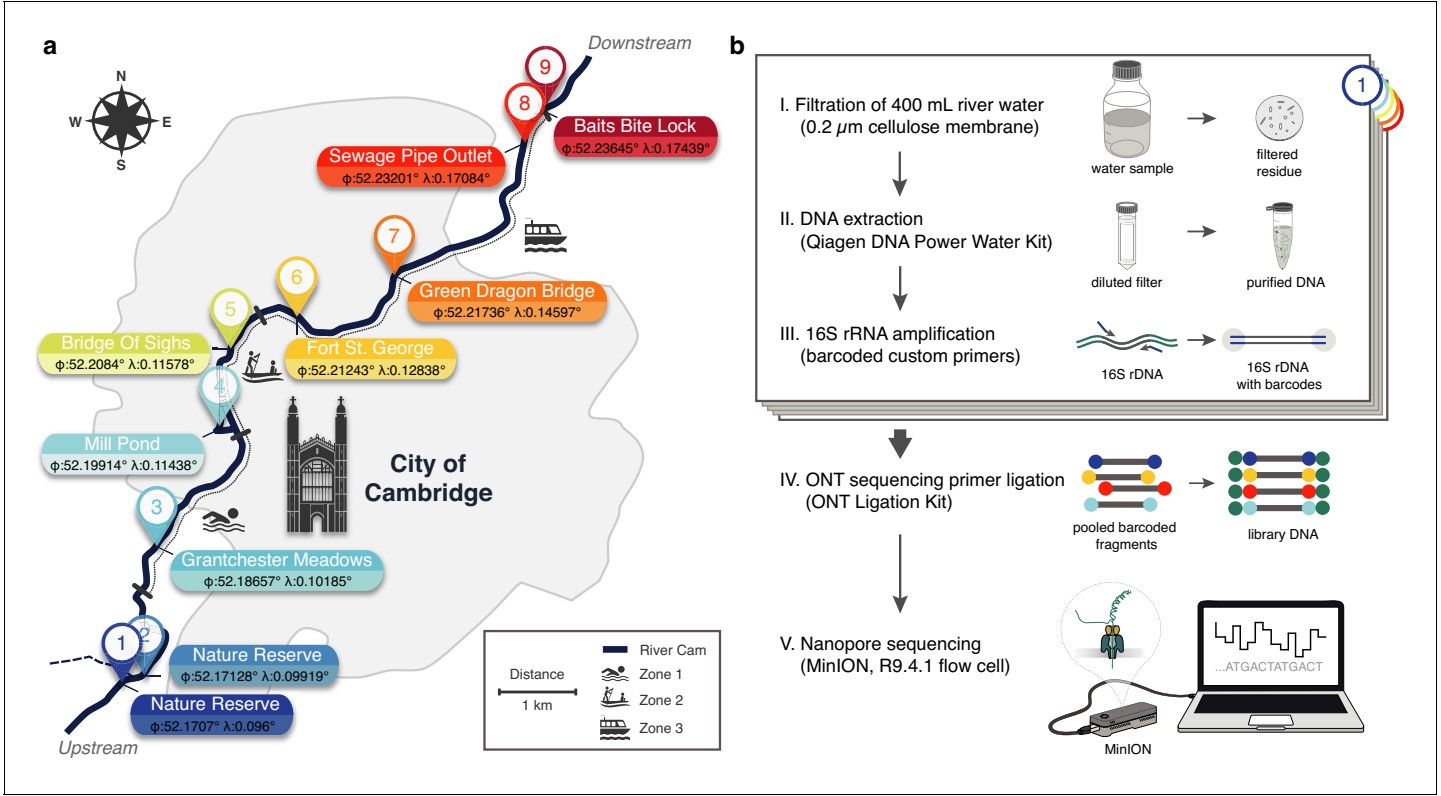

**Figure 1.** Freshwater microbiome study design and experimental setup. (a) Schematic map of Cambridge (UK), illustrating sampling locations (colour-coded) along the River Cam. Geographic coordinates of latitude and longitude are expressed as decimal fractions according to the global positioning system. (b) Laboratory workflow to monitor bacterial communities from freshwater samples using nanopore sequencing (Materials and methods). The online version of this article includes the following figure supplement(s) for figure 1:

**Figure supplement 1.** Bioinformatics consensus workflow.

concerns about the technology's base-level accuracy, mobile MinION setups have already been transformative for real-time tracing and rapid data sharing during bacterial and viral pathogen outbreaks (*Boykin et al., 2019*; *Chan et al., 2020*; *Faria et al., 2018*; *Faria et al., 2017*; *Kafetzopoulou et al., 2019*; *Quick et al., 2015*; *Quick et al., 2016*). In the context of freshwater analysis, a MinION whole-genome shotgun sequencing protocol has recently been leveraged for a comparative study of 11 rivers (*Reddington et al., 2020*). This report highlights key challenges which emerge in serial monitoring scenarios of a relatively low-input DNA substrate (freshwater), for example large sampling volumes (2–4 l) and small shotgun fragments (mean < 4 kbp). We reasoned that targeted DNA amplification may be a suitable means to bypass these bottlenecks and assess river microbiomes with nanopore sequencing.

Here, we report a simple, cost-effective workflow to assess and monitor microbial freshwater ecosystems with targeted nanopore DNA sequencing. Our benchmarking study involves the design and optimisation of essential experimental steps for multiplexed MinION usage in the context of local environments, together with an evaluation of computational methods for the bacterial classification of nanopore sequencing reads from metagenomic libraries. To showcase the resolution of sequencing-based aquatic monitoring in a spatiotemporal setting, we combine DNA analyses with physico-chemical measurements of surface water samples collected at nine locations within a confined ~12 km reach of the River Cam passing through the city of Cambridge (UK) in April, June, and August 2018.

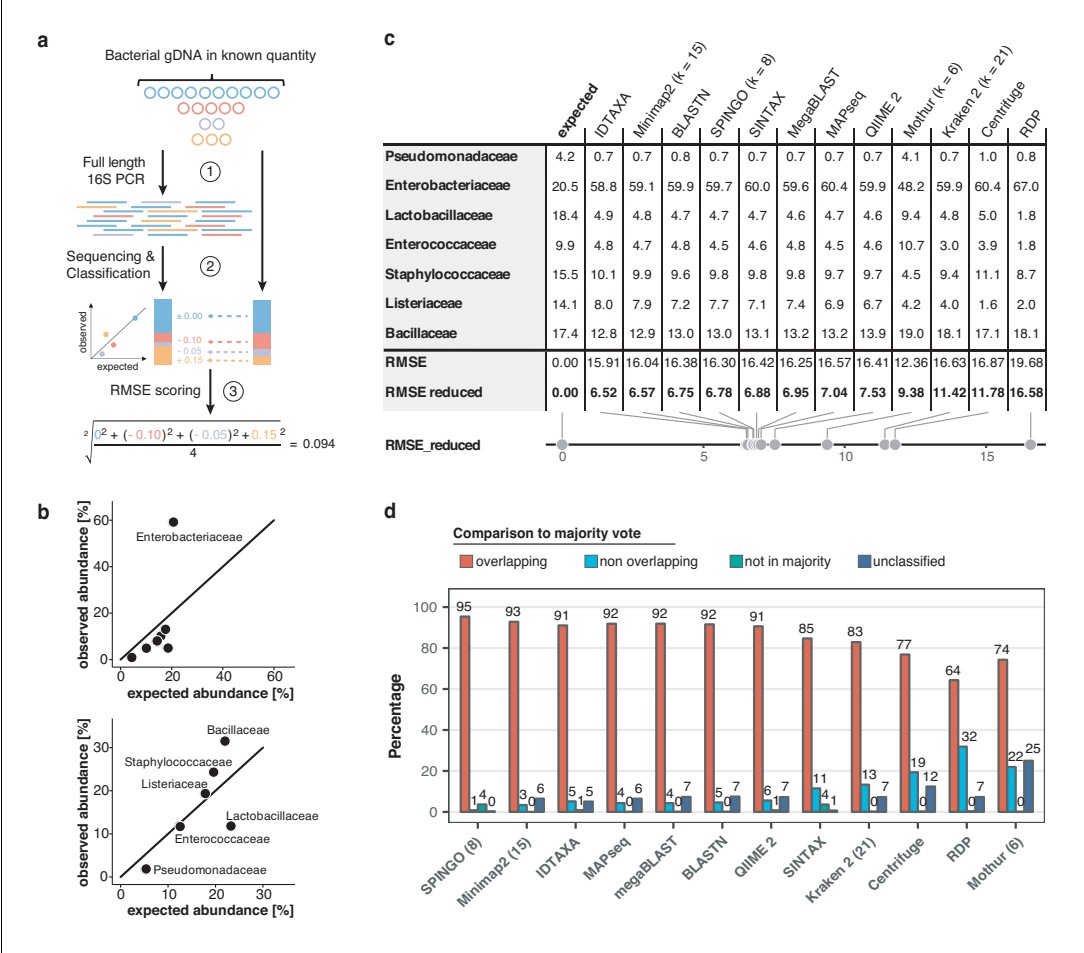

**Figure 2.** Benchmarking of classification tools with nanopore full-length 16S sequences. (**a**) Schematic of mock community quantification performance testing. (**b**) Observed vs. expected read fraction of bacterial families present in 10,000 nanopore reads randomly drawn from mock community sequencing data. Example representation of Minimap2 (kmer length 15) quantifications with (upper) and without (lower) *Enterobacteriaceae* (Materials and methods). (**c**) Mock community classification output summary for twelve classification tools tested against the same 10,000 reads. Root mean squared errors observed and expected bacterial read fractions are provided with (RMSE) and without *Enterobacteriaceae* (RMSE reduced). (**d**) Classification output summary for 10,000 reads randomly drawn from an example freshwater sample (Materials and methods). 'Overlapping' fractions (red) represent agreements of a classification tool with the majority of tested methods on the same reads, while 'non-overlapping' fractions (light blue) represent disagreements. Dark green sets highlight rare taxon assignments not featured in any of the 10,000 majority classifications, while dark blue bars show unclassified read fractions.

The online version of this article includes the following figure supplement(s) for figure 2:

**Figure supplement 1.** Benchmarking of twelve taxonomic classifiers with nanopore full-length 16S sequences.

**Figure supplement 2.** Key challenges of freshwater monitoring with nanopore sequencing.

## Results

### Experimental design and computational workflows

Using a bespoke workflow, nanopore full-length (V1-V9) 16S ribosomal RNA (rRNA) gene sequencing was performed on all location-barcoded freshwater samples at each of the three time points (*Figure 1*; *Supplementary file 1*; Materials and methods). River isolates were multiplexed with negative controls (deionised water) and mock community controls composed of eight bacterial species in known mixture proportions.

To obtain valid taxonomic assignments from freshwater sequencing profiles using nanopore sequencing, twelve different classification tools were compared through several performance metrics (*Figure 2*; *Figure 2—figure supplement 1*; Materials and methods). Our comparison included

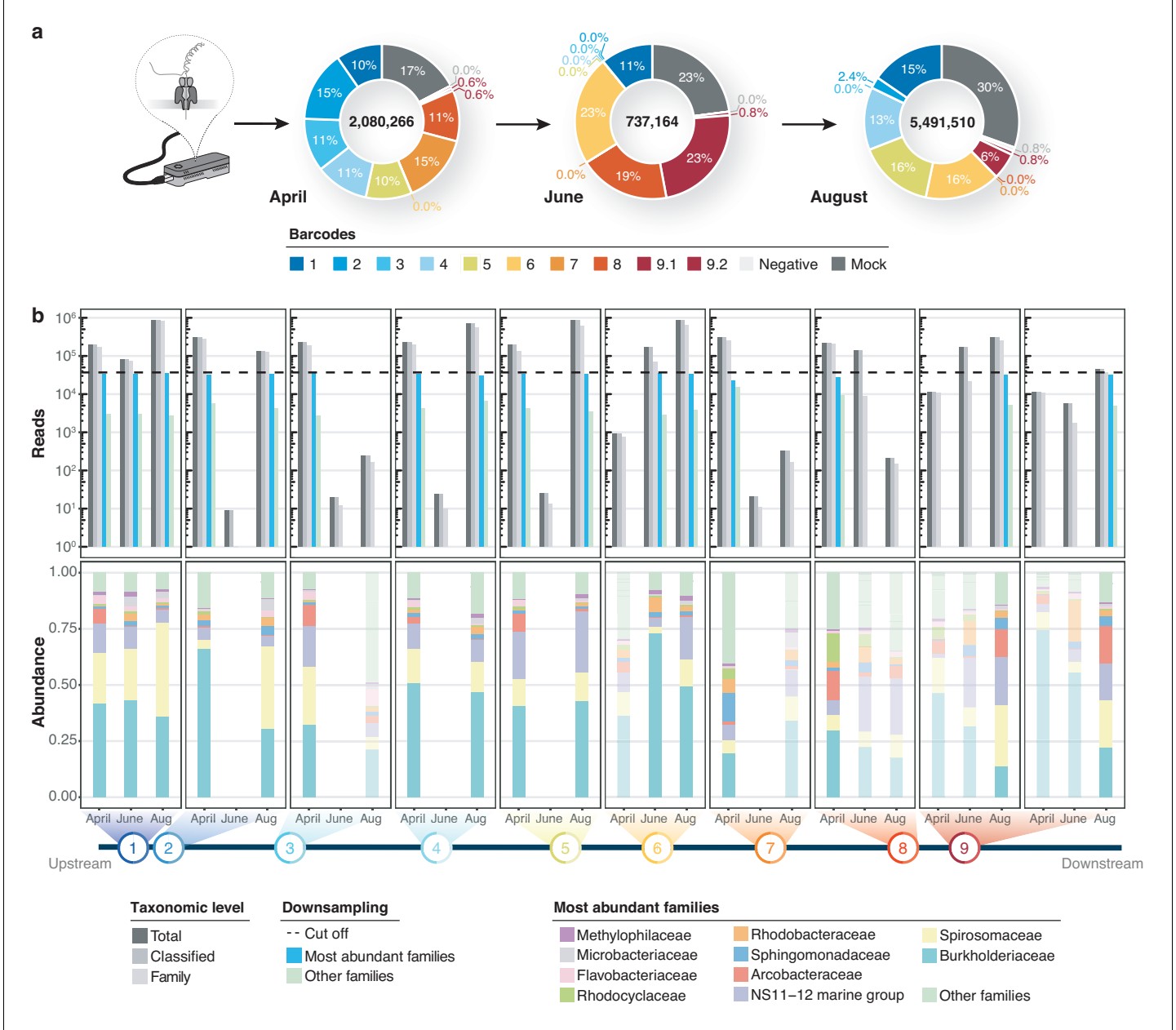

**Figure 3.** Bacterial diversity of the River Cam. (**a**) Nanopore sequencing output summary. Values in the centre of the pie charts depict total numbers of classified nanopore sequences per time point. Percentages illustrate representational fractions of locations and control barcodes (negative control and mock community). (**b**) Read depth and bacterial classification summary. Upper bar plot shows the total number of reads, and the number of reads classified to any taxonomic level, to at least bacterial family level, to the ten most abundant bacterial families across all samples, or to other families. Rarefaction cut-off displayed at 37,000 reads (dashed line). Lower bar plot features fractions of the ten most abundant bacterial families across the samples with more than 100 reads. Colours in bars for samples with less than 37,000 reads are set to transparent.

The online version of this article includes the following figure supplement(s) for figure 3:

**Figure supplement 1.** Impact of rarefaction on diversity estimation.

established classifiers such as RDP (*Wang et al., 2007*), Kraken (*Wood and Salzberg, 2014*), and Centrifuge (*Kim et al., 2016*), as well as more recently developed methods optimised for higher sequencing error rates such as IDTAXA (*Murali et al., 2018*) and Minimap2 (*Li, 2018*). An *Enterobacteriaceae* overrepresentation was observed across all replicates and classification methods, pointing towards a consistent *Escherichia coli* amplification bias potentially caused by skewed taxonomic

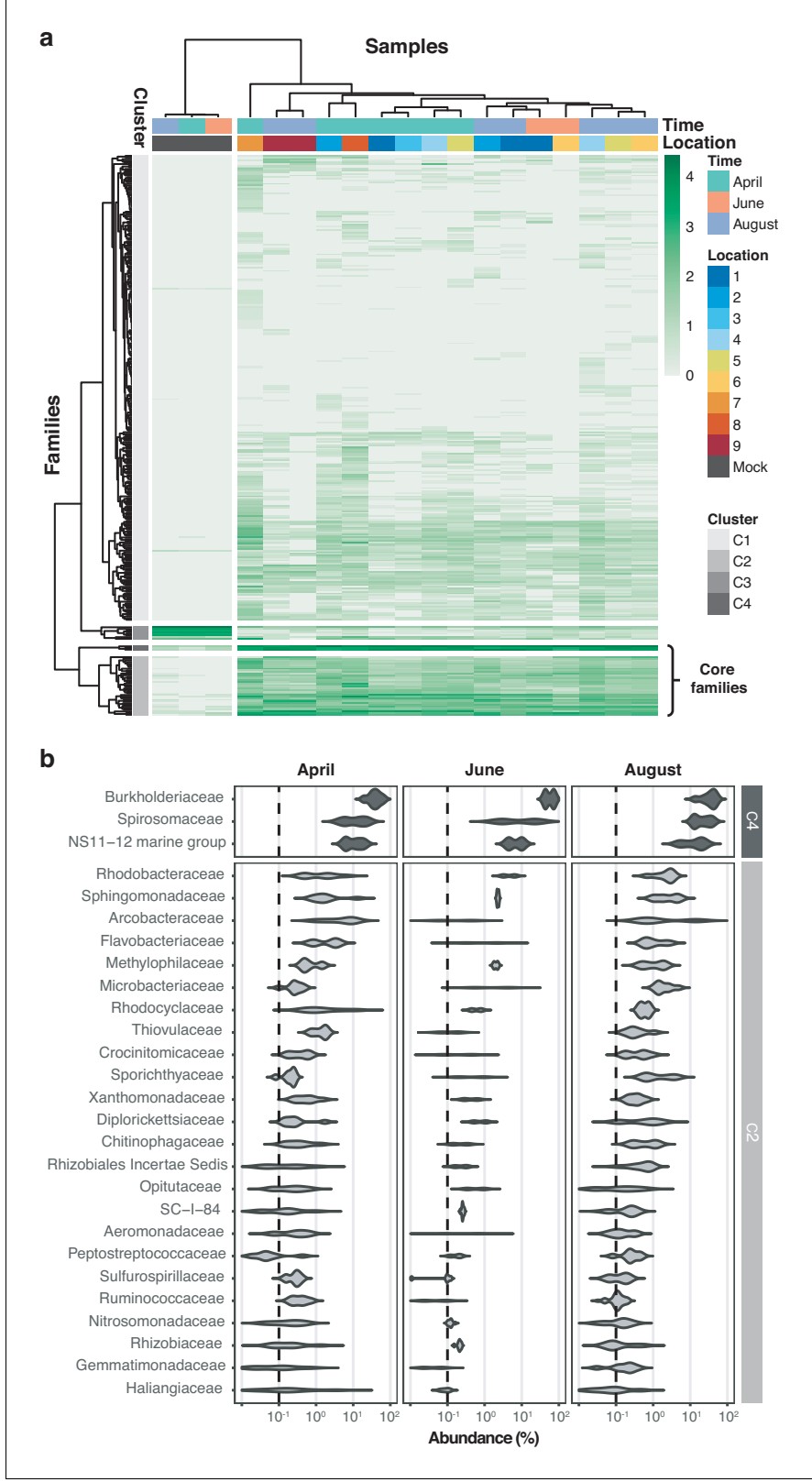

**Figure 4.** Core microbiome of the River Cam. (**a**) Hierarchical clustering of bacterial family abundances across freshwater samples after rarefaction, together with the mock community control. Four major clusters of bacterial families occur, with two of these (C2 and C4) corresponding to the core microbiome of ubiquitously abundant families, one (C3) corresponding to the main mock community families and one (C1) corresponding to the majority

*Figure 4 continued on next page*

*Figure 4 continued*

of rare accessory taxa. (**b**) Detailed river core microbiome. Violin plots summarise fractional representation of bacterial families from clusters C2 and C4 (log$_{10}$ scale of relative abundance [%] across all samples, n$_{April}$ = 7, n$_{June}$ = 2, n$_{August}$ = 7), sorted by median total abundance. Vertical dashed lines depict 0.1% proportion.

The online version of this article includes the following figure supplement(s) for figure 4:

**Figure supplement 1.** River Cam core microbiome analysis on the bacterial genus level.

specificities of the selected 16S primer pair 27F and 1492R (*Frank et al., 2008*; *Figure 2b*). Root mean square errors (RMSE) between observed and expected bacteria of the mock community differed slightly across all classifiers (*Figure 2c*). Robust quantifications were obtained by Minimap2 alignments against the SILVA v.132 database (*Quast et al., 2013*), for which 99.68% of classified reads aligned to the expected mock community taxa (mean sequencing accuracy 92.08%; *Figure 2—figure supplement 2c*). Minimap2 classifications reached the second lowest RMSE (excluding *Enterobacteriaceae*), and relative quantifications were highly consistent between mock community replicates. Benchmarking of the classification tools on one aquatic sample further confirmed Minimap2's reliable performance in a complex bacterial community (*Figure 2d*), although other tools such as MAPseq (*Matias Rodrigues et al., 2017*), SPINGO (*Allard et al., 2015*), or IDTAXA also produced highly concordant results – despite variations in memory usage and runtime over several orders of magnitude (*Figure 2—figure supplement 1b*).

## Diversity analysis and river core microbiome

Using Minimap2 classifications within our bioinformatics consensus workflow (*Figure 1—figure supplement 1*; Materials and methods), we then inspected sequencing profiles of three independent MinION runs for a total of 30 river DNA isolates and six controls. This yielded ~8.3 million sequences with exclusive barcode assignments (*Figure 3a*; *Supplementary file 2*). Overall, 82.9% (n = 6,886,232) of raw reads could be taxonomically assigned to the family level (*Figure 3b*). To account for variations in sample sequencing depth, rarefaction with a cut-off at 37,000 reads was applied to all samples. While preserving ~90% of the original family level taxon richness (Mantel test, R = 0.814, p = 2.1*10$^{-4}$; *Figure 3—figure supplement 1a–b*), this conservative thresholding resulted in the exclusion of 14 samples, mostly from the June time point, for subsequent high-resolution analyses. The 16 remaining surface water samples revealed moderate levels of microbial heterogeneity (*Figure 3b*; *Figure 3—figure supplement 1c*): microbial family alpha diversity ranged between 0.46 (June-6) and 0.92 (April-7) (Simpson index), indicating low-level evenness with a few taxonomic families that account for the majority of the metagenomic signal.

Hierarchical clustering of taxon profiles showed a dominant core microbiome across all aquatic samples (clusters C2 and C4, *Figure 4a*). The most common bacterial families observed were *Burkholderiaceae* (40.0%), *Spirosomaceae* (17.7%), and NS11-12 marine group (12.5%), followed by *Arcobacteraceae* (4.8%), *Sphingomonadaceae* (2.9%), and *Rhodobacteraceae* (2.5%) (*Figure 4b*). Members of these families are commonly associated with aquatic environments; for example, major fractions of *Burkholderiaceae* reads originated from genera such as *Limnohabitans*, *Rhodoferax*, *Polynucleobacter*, or *Aquabacterium* (*Figure 4—figure supplement 1*), which validates the suitability of this nanopore metagenomics workflow. Hierarchical clustering additionally showed that two biological replicates collected at the same location and time point (April samples 9.1 and 9.2), grouped with high concordance; this indicates that spatiotemporal trends are discernible even within a highly localised context.

Besides the dominant core microbiome, microbial profiles showed a marked arrangement of time dependence, with water samples from April grouping more distantly to those from June and August. Principal component analysis (PCA) illustrates the seasonal divergence among the three sampling months (*Figure 5a*; *Figure 5—figure supplement 1*). The strongest differential abundances along the seasonal axis of variation (PC3) derived from *Carnobacteriaceae* (*Figure 5b*), a trend also highlighted by taxon-specific log-normal mixture model decomposition between the two seasons (April vs. June/August; p < 0.01; Materials and methods). Indeed, members of this bacterial family have been primarily isolated from cold substrates (*Lawson and Caldwell, 2014*).

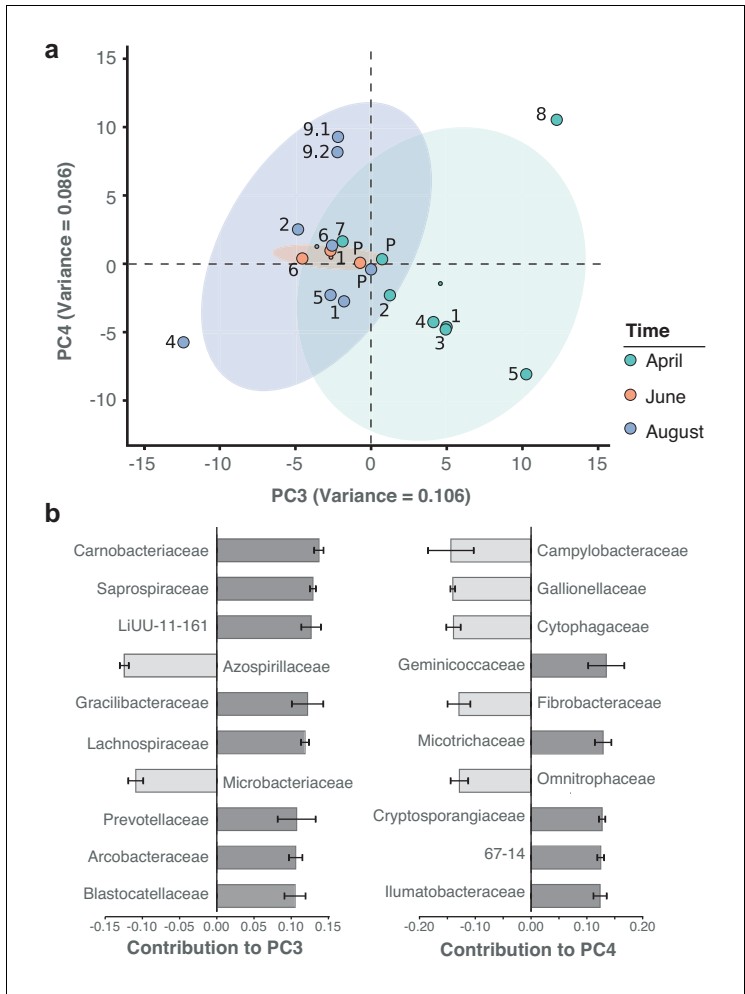

**Figure 5.** Spatiotemporal axes of taxonomic diversity in the River Cam. (a) PCA of bacterial composition across locations, indicating community dissimilarities along the main time (PC3) and spatial (PC4) axes of variation; dots coloured according to time points. Kruskal-Wallis test on PC3 component contributions, with post-hoc Mann-Whitney U rank test (April vs. August): $p = 2.2*10^{-3}$. (b) Contribution of individual bacterial families to the PCs in (a). Error bars represent the standard deviation of these families across four independent rarefactions. The online version of this article includes the following figure supplement(s) for figure 5:

**Figure supplement 1.** Principal component analysis of river bacterial family compositions.

## Hydrochemistry and seasonal profile of the River Cam

While a seasonal difference in bacterial composition can be expected due to increasing water temperatures in the summer months, additional changes may have also been caused by alterations in river hydrochemistry and flow rate (*Figure 6a*; *Figure 6—figure supplement 1*; *Supplementary file 1*). To assess this effect in detail, we measured the pH and a range of major and trace cations in all river water samples using inductively coupled plasma-optical emission spectroscopy (ICP-OES), as well as major anions using ion chromatography (Materials and methods). As with the bacterial composition dynamics, we observed significant temporal variation in water chemistry, superimposed on a spatial gradient of generally increasing sodium and chloride concentrations along the river reach (*Figure 6b–c*). This spatially consistent effect is likely attributed to wastewater and agricultural discharge inputs in and around Cambridge city. A comparison of the major element chemistry in the River Cam transect with the world's 60 largest rivers further corroborates the likely impact of anthropogenic pollution in this fluvial ecosystem (*Gaillardet et al., 1999*; *Figure 6d*; Materials and methods).

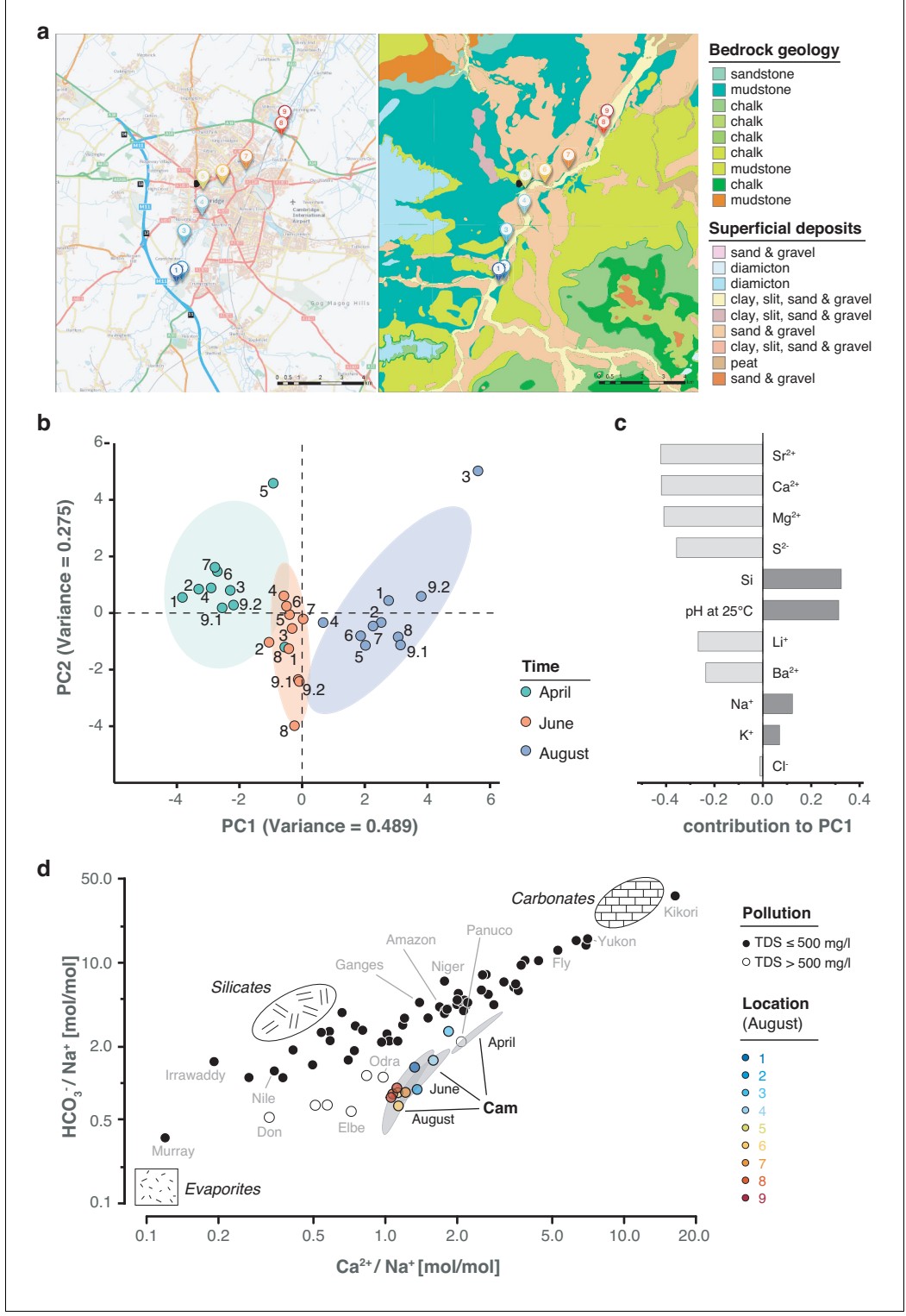

**Figure 6.** Geological and hydrochemical profile of the River Cam and its basin. (a) Outline of the Cam River catchment surrounding Cambridge (UK), and its corresponding lithology. Overlay of bedrock geology and superficial deposits (British Geological Survey data: DiGMapGB-50, 1:50,000 scale) is shown as visualised by GeoIndex. Bedrock is mostly composed of subtypes of Cretaceous limestone (chalk), gault (clay, sand), and mudstone. Approximate sampling locations are colour-coded as in *Figure 1*. (b) Principal component analysis of measured pH and 13 inorganic solute concentrations of this study's 30 river surface water samples. PC1 (~49% variance) displays a strong, continuous temporal shift in hydrochemistry. (c) Parameter contributions to PC1 in (b),

*Figure 6 continued on next page*

*Figure 6 continued*

highlighting a reduction in water hardness (Ca$^{2+}$, Mg$^{2+}$) and increase in pH towards the summer months (June and August). (**d**) Mixing diagram with Na$^+$-normalised molar ratios, representing inorganic chemistry loads of the world's 60 largest rivers; open circles represent polluted rivers with total dissolved solid (TDS) concentrations > 500 mg l$^{-1}$. Cam River ratios are superimposed as ellipses from ten samples per month (50% confidence, respectively). Separate data points for all samples from August are also shown and colour-coded, indicating the upstream-to-downstream trend of Na$^+$ increase (also observed in April and June). End-member signatures show typical chemistry of small rivers draining these lithologies exclusively (carbonate, silicate and evaporite).

The online version of this article includes the following figure supplement(s) for figure 6:

**Figure supplement 1.** Cambridge weather and River Cam flow rate.

## Maps of potential bacterial pathogens at species level resolution

Freshwater sources throughout the United Kingdom have been notorious for causing bacterial infections such as leptospirosis (*Public Health England, 2016*; *Public Health England, 2019*). In line with the physicochemical profile of the River Cam, we therefore next determined the spatiotemporal enrichment of potentially important functional bacterial taxa through nanopore sequencing. We retrieved 55 potentially pathogenic bacterial genera through integration of species known to affect human health (*Jin et al., 2018*; *Wattam et al., 2017*), and also 13 wastewater-associated bacterial genera (*Global Water Microbiome Consortium et al., 2019*; *Supplementary file 3*). Of these, 21 potentially pathogenic and 8 wastewater-associated genera were detected across all of the river samples (*Figure 7*; Materials and methods). Many of these signals were stronger downstream of urban sections, within the mooring zone for recreational and residential barges (location 7; *Figure 1a*) and in the vicinity of sewage outflow from a nearby wastewater treatment plant (location 8). The most prolific candidate pathogen genus observed was *Arcobacter*, which features multiple species implicated in acute gastrointestinal infections (*Kayman et al., 2012*).

In general, much of the taxonomic variation across all samples was caused by sample April-7 (PC1 explains 27.6% of the overall variance in bacterial composition; *Figure 5—figure supplement 1a–b*). Its profile was characterised by an unusual dominance of *Caedibacteraceae*, *Halomonadaceae* and others (*Figure 5—figure supplement 1c*). Isolate April-8 also showed a highly distinct bacterial composition, with some families nearly exclusively occurring in this sample (outlier analysis; Materials and methods). The most predominant bacteria in this sewage pipe outflow are typically found in wastewater sludge or have been shown to contribute to nutrient pollution from effluents of wastewater plants, such as *Haliangiaceae*, *Nitospiraceae*, *Rhodocyclaceae*, and *Saprospiracea* (*Nielsen et al., 2012*; *Global Water Microbiome Consortium et al., 2019*; *Figure 7*).

Using multiple sequence alignments between nanopore reads and pathogenic species references, we further resolved the phylogenies of three common potentially pathogenic genera occurring in our river samples, *Legionella*, *Salmonella*, and *Pseudomonas* (*Figure 8a–c*; Materials and methods). While *Legionella* and *Salmonella* diversities presented negligible levels of known harmful species, a cluster of reads in downstream sections indicated a low abundance of the opportunistic, environmental pathogen *Pseudomonas aeruginosa* (*Figure 8c*).

Along the course here investigated, we also found significant variations in relative abundances of the *Leptospira* genus, which was recently described to be enriched in wastewater effluents in Germany (*Numberger et al., 2019*; *Figure 8d*). Indeed, the peak of River Cam *Leptospira* reads fell into an area of increased sewage influx (~0.1% relative abundance; *Figure 7*). The *Leptospira* genus contains several potentially pathogenic species capable of causing life-threatening leptospirosis through waterborne infections, however, also features close-related saprophytic and 'intermediate' taxa (*Vincent et al., 2019*; *Wynwood et al., 2014*). To resolve its complex phylogeny in the River Cam surface, we aligned *Leptospira* reads from all samples together with many reference sequences assigned to pre-classified pathogenic, saprophytic and other environmental *Leptospira* species (*Figure 8d*; *Supplementary file 4*; Materials and methods). Despite the presence of nanopore sequencing errors (*Figure 2—figure supplement 2c*) and correspondingly inflated read divergence, we could pinpoint spatial clusters and a distinctly higher similarity between our amplicons and saprophytic rather than pathogenic *Leptospira* species. These findings were subsequently validated by

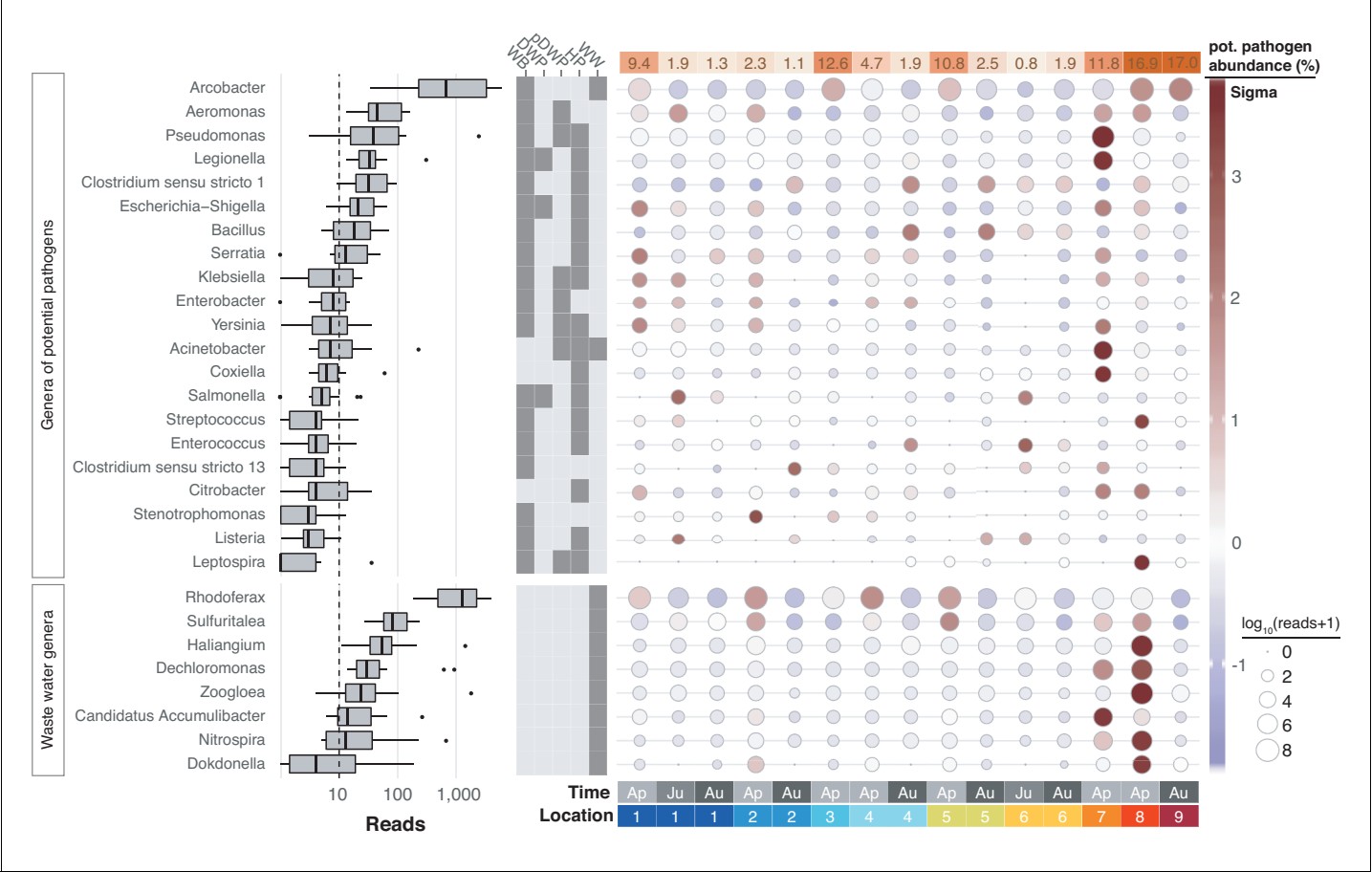

**Figure 7.** Potentially pathogenic and wastewater treatment related bacteria in the River Cam. Boxplots on the left show the abundance distribution across locations per bacterial genus. Error bars represent Q1 – 1.5*IQR (lower), and Q3 + 1.5*IQR (upper), respectively; Q1: first quartile, Q3: third quartile, IQR: interquartile range. The central table depicts the categorisation of subsets of genera as waterborne bacterial pathogens (WB), drinking water pathogens (DWP), potential drinking water pathogens (pDWP), human pathogens (HP), and core genera from wastewater treatment plants (WW) (dark grey: included, light grey: excluded) (**Supplementary file 3**). The right-hand circle plot shows the distribution of bacterial genera across locations of the River Cam. Circle sizes represent overall read size fractions, while circle colours (sigma scheme) represent the standard deviation from the observed mean relative abundance within each genus.

targeted, *Leptospira* species-specific qPCR (**Supplementary file 5**; Materials and methods), confirming that R9.4.1 nanopore sequencing quality is already high enough to yield indicative results for bacterial monitoring workflows at the species level.

## Discussion

Using a cost-effective, easily adaptable and scalable framework, we provide the first spatiotemporal nanopore sequencing atlas of bacterial microbiota throughout the course of a river. Our results suggest that this workflow allows for robust assessments of both, the core microbiome of an example fluvial ecosystem and heterogeneous bacterial compositions in the context of supporting physical (temperature, flow rate) and hydrochemical (pH, inorganic solutes) parameters. We show that the technology's current sequencing accuracy of ~92% allows for the designation of significant human pathogen community shifts along rural-to-urban river transitions, as illustrated by downstream increases in the abundance of pathogen candidates.

Our assessment of bioinformatics workflows for taxonomic classification highlights current challenges with error-prone nanopore sequences. A number of recent reports feature bespoke 16S read classification schemes centred around a single software (*Acharya et al., 2019*; *Benítez-Páez et al., 2016*; *Kerkhof et al., 2017*; *Nygaard et al., 2020*), and others integrated outputs from two

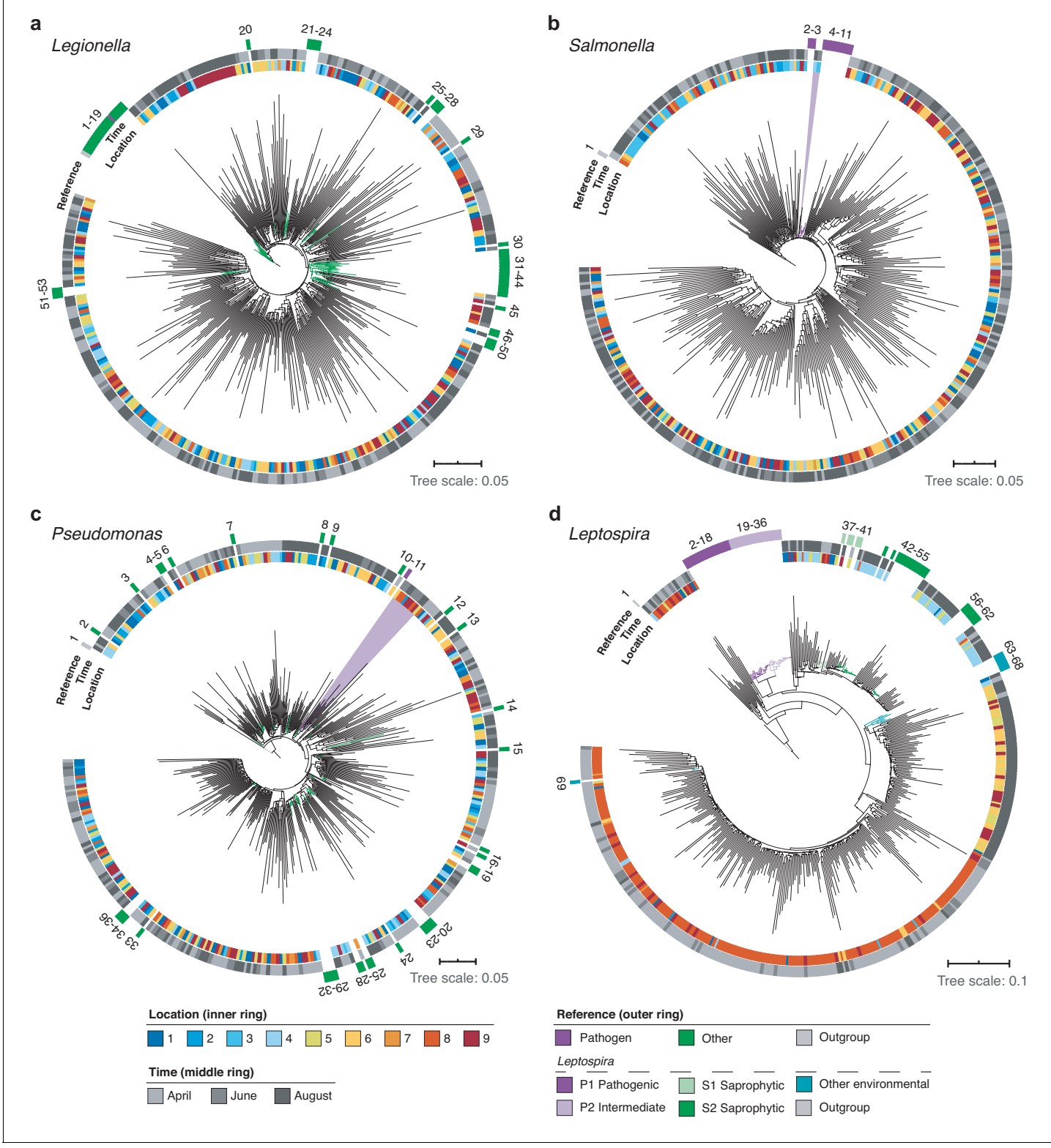

**Figure 8.** High-resolution phylogenetic clustering of candidate pathogenic genera in the River Cam. Phylogenetic trees illustrating multiple sequence alignments of exemplary River Cam nanopore reads (black branches) classified as (a) *Legionella*, (b) *Salmonella*, (c) *Pseudomonas*, or (d) *Leptospira*, together with known reference species sequences ranging from pathogenic to saprophytic taxa within the same genus (coloured branches). Reference species sequences are numbered in clockwise orientation around the tree (*Supplementary file 4*). Nanopore reads highlighted in light violet background display close clustering with pathogenic isolates of (b) *Salmonella spp.* and (c) *Pseudomonas aeruginosa*.

methods (*Cuscó et al., 2018*). Through systematic benchmarking of twelve different classification tools, using matched mock community and river water datasets with respect to the SILVA v.132 reference database, we lay open key differences in terms of these methods' read (mis)classification rates, consensus agreements, speed and memory performance metrics. For example, our results indicate that very fast implementations like Kraken 2 or Centrifuge yield less accurate classifications than slightly slower and more memory-demanding frameworks such as Minimap2 (*Figure 2*; *Figure 2—figure supplement 1*).

Using Minimap2, 16.2% of freshwater-derived sequencing reads were assigned to a bacterial species on average, thereby primarily encouraging automated analyses on the genus (65.6% assigned) or family level (76.6% assigned). As nanopore sequencing quality continues to increase through refined pore chemistries, basecalling algorithms and consensus sequencing workflows (*Calus et al., 2018*; *Karst et al., 2021*; *Latorre-Pérez et al., 2020*; *Rang et al., 2018*; *Santos et al., 2020*; *Zurek et al., 2020*), future bacterial taxonomic classifications are likely to improve and advance opportunities for species discovery.

We show that nanopore amplicon sequencing data can resolve the core microbiome of a freshwater body, as well as its temporal and spatial fluctuations. Common freshwater bacteria account for the vast majority of taxa in the River Cam; this includes *Sphingomonadaceae*, which had also been previously found at high abundance in source water from the same river (*Rowe et al., 2016*). Our findings suggest that the differential abundances of *Carnobacteriaceae* most strongly contribute to seasonal loadings in the River Cam. *Carnobacteriaceae* have been previously associated with a range of low-temperature environments (*Lawson and Caldwell, 2014*), and we found these taxa to be more abundant in colder April samples (mean 11.3°C, vs. 15.8°C in June and 19.1°C in August). This might help to further establish this family as an indicator for bacterial community shifts along with temperature fluctuations, albeit the influence of co-occurring hydrochemical trends such as water hardness, dissolved carbon or flow speed changes should also be noted (*Figure 6b–d*; *Figure 6—figure supplement 1d*).

Most routine freshwater surveillance frameworks focus on semi-quantitative diagnostics of only a limited number of target taxa, such as pathogenic *Salmonella*, *Legionella* and faecal coliforms (*Ramírez-Castillo et al., 2015*; *Tan et al., 2015*), whereas metagenomics approaches can give a complete and detailed overview of environmental microbial diversity. Besides nanopore shotgun-sequencing (*Reddington et al., 2020*), our proof-of-principle analysis highlights that targeted full-length 16S rRNA gene MinION sequencing is a suitable complement to hydrochemical controls in pinpointing relatively contaminated freshwater sites, some of which in case of the River Cam had been previously highlighted for their pathogen diversity and abundance of antimicrobial resistance genes (*Rowe et al., 2017*; *Rowe et al., 2016*). Nanopore amplicon sequencing has here allowed us to reliably distinguish closely related pathogenic and non-pathogenic bacterial species of the common *Legionella*, *Salmonella*, *Pseudomonas*, and *Leptospira* genera. For *Leptospira* bacteria, which are of particular interest to communal stakeholders of the River Cam, we validated nanopore sequencing results through the gold standard qPCR workflow of Public Health England (*Supplementary file 5*). In order to also study the potential viability and functional implications of sequenced pathogen candidates for public health, we encourage future studies to combine nanopore based freshwater metagenomics with targeted follow-up measurements of living pathogens by established microbiological approaches, including species-specific isolation and subsequent culturing.

A number of experimental intricacies should be addressed towards nanopore freshwater sequencing with our approach, mostly by scrutinising water DNA extraction yields, PCR biases and molar imbalances in barcode multiplexing (*Figure 3a*; *Figure 2—figure supplement 2a–b*; *Supplementary file 2*). Similar to challenges with other organic substrates, microbial raw DNA extraction protocols require careful pre-testing and optimisation towards the physicochemical composition of a given freshwater source, in order to avoid both taxonomic enrichment biases and drop-offs in total yield. One example lies in the optimisation of the filtrate volume – in this study, membrane DNA extraction from 400 mL River Cam water was sufficient to yield valuable insights, while as much as 10,000 mL were used in a previous study of the same river (*Rowe et al., 2016*). Moreover, potentially dissolved inhibitory compounds for DNA extraction, sample cooling and storage chains should be thoroughly considered for larger and remote river monitoring projects. We witnessed that yield variations may bear negative effects on the molar balance of barcoded nanopore sequencing runs, as illustrated by elevated sample dropouts in June 2018, emphasising the need for

highly accurate concentration measurement and scaling when dozens of input DNA sources are pooled. Our study further highlights that MinION (R9.4.1) flow cell throughput can fluctuate by an order of magnitude, altogether causing the exclusion of measurements upon application of a conservative read threshold. We reason that real-time selective nanopore sequencing could serve as a powerful means to improve barcode balances in the context of multiplexed 16S analyses (*Loose et al., 2016*), albeit such approaches are yet undergoing computational optimisations (*Kovaka et al., 2020*; *Payne et al., 2020*).

Our results show that it would already be theoretically feasible to obtain meaningful river microbiota from > 100 barcoded samples on a single nanopore flow cell, thereby enabling water monitoring projects involving large collections at costs below £20 per sample (*Supplementary file 6*). In line with this, ONT has already released several commercial 96-barcode multiplexing kits for PCR- and non-PCR-based applications, as well as the smaller 'Flongle' flow cell with considerably reduced cost as compared to the traditional MinION model. On the other hand, shotgun nanopore sequencing approaches may bypass pitfalls associated with amplicon sequencing, namely taxon-specific primer biases (*Frank et al., 2008*), 16S rRNA copy number fluctuations between species (*Darby et al., 2013*) or the omission of functionally relevant sequence elements. In combination with sampling protocol adjustments, shotgun nanopore sequencing could moreover be used for the serial monitoring of eukaryotic microorganisms and viruses in freshwater ecosystems (*Reddington et al., 2020*).

Since the commercial launch of the MinION in 2015, a wide set of microbial nanopore sequencing applications in the context of rRNA gene (*Benítez-Páez et al., 2016*; *Cuscó et al., 2018*; *Kerkhof et al., 2017*; *Nygaard et al., 2020*) and shotgun (*Leggett et al., 2020*; *Nicholls et al., 2019*; *Reddington et al., 2020*; *Stewart et al., 2019*) metagenomics have attracted the interest of a growing user community. Two independent case studies have recently provided decomposition analyses of faecal bacterial pathogens in MinION libraries derived from river and spring waters in Montana, USA (*Hamner et al., 2019*) and Kathmandu Valley, Nepal (*Acharya et al., 2019*). Although it is to be expected that short-read metagenomics technology continues to provide valuable environmental insights, as illustrated through global cataloguing efforts of ocean (*Tara Oceans coordinators et al., 2015*) and wastewater (*Global Water Microbiome Consortium et al., 2019*) microbiomes, due to their large sizes and fixed costs these traditional platforms remain unfeasible for the monitoring of remote environments – especially in low-resource settings. We reason that the convenience of MinION handling and complementary development of portable DNA purification methods (*Boykin et al., 2019*; *Gowers et al., 2019*) will allow for such endeavours to become increasingly accessible to citizens and public health organisations around the world, ultimately democratising the opportunities and benefits of DNA sequencing.

## Materials and methods

### Freshwater sampling

We monitored nine distinct locations along a 11.62 km reach of the River Cam, featuring sites upstream, downstream and within the urban belt of the city of Cambridge, UK. Measurements were taken at three time points, in two-month intervals between April and August 2018 (*Figure 1*; *Supplementary file 1a*). To warrant river base flow conditions and minimise rain-derived biases, a minimum dry weather time span of 48 hr was maintained prior to sampling (*Fisher et al., 2015*). One litre of surface water was collected in autoclaved DURAN bottles (Thermo Fisher Scientific, Waltham, MA, USA), and cooled to 4°C within 3 hr. Two bottles of water were collected consecutively for each time point, serving as biological replicates of location 9 (samples 9.1 and 9.2).

### Physical and chemical metadata

We assessed various chemical, geological and physical properties of the River Cam (*Figure 6*; *Figure 6—figure supplement 1*; *Supplementary file 1b-c*).

In situ water temperature was measured immediately after sampling. To this end, we linked a DS18B20 digital temperature sensor to a portable custom-built, grid mounted Arduino nano v3.0 system. The pH was later recorded under temperature-controlled laboratory conditions, using a pH edge electrode (HI-11311, Hanna Instruments, Woodsocket, RI, USA).

To assess the dissolved ion concentrations in all collected water samples, we aerated the samples for 30 s and filtered them individually through a 0.22 µM pore-sized Millex-GP polyethersulfone syringe filter (MilliporeSigma, Burlington, MA, USA). Samples were then acidified to pH ~2, by adding 20 µL of 7M distilled $HNO_3$ per 3 mL sample. Inductively coupled plasma-optical emission spectroscopy (ICP-OES, Agilent 5100 SVDV; Agilent Technologies, Santa Clara, CA, USA) was used to analyse the dissolved cations $Na^+$, $K^+$, $Ca^{2+}$, $Mg^{2+}$, $Ba^{2+}$, $Li^+$, as well as Si and $SO_4^{2-}$ (as total S) (*Supplementary file 1b*). International water reference materials (SLRS-5 and SPS-SW2) were interspersed with the samples, reproducing certified values within 10% for all analysed elements. Chloride concentrations were separately measured on 1 mL of non-acidified aliquots of the same samples, using a Dionex ICS-3000 ion chromatograph (Thermo Fisher Scientific, Waltham, MA, USA) (*Supplementary file 1b*). Long-term repeat measurements of a USGS natural river water standard T-143 indicated precision of more than 4% for $Cl^-$. However, the high $Cl^-$ concentrations of the samples in this study were not fully bracketed by the calibration curve and we therefore assigned a more conservative uncertainty of 10% to $Cl^-$ concentrations.

High calcium and magnesium concentrations were recorded across all samples, in line with hard groundwater and natural weathering of the Cretaceous limestone bedrock underlying the river catchment (*Figure 6a*). There are no known evaporite salt deposits in the river catchment, and therefore the high dissolved $Na^+$, $K^+$, and $Cl^-$ concentrations in the River Cam are likely derived from anthropogenic inputs (*Rose, 2007*; *Figure 6c–d*). We calculated bicarbonate concentrations through a charge balance equation (concentrations in mol/L): conc ($HCO_3^-$) = conc ($Li^+$) + conc ($Na^+$) + conc ($K^+$) + 2*conc ($Mg^{2+}$) + 2*conc ($Ca^{2+}$) - conc ($Cl^-$) - 2*conc ($S^{2-}$).

The total dissolved solid (TDS) concentration across the 30 freshwater samples had a mean of 458 mg/L (range 325–605 mg/L) which is relatively high compared to most rivers, due to (1) substantial solute load in the Chalk groundwater (particularly $Ca^{2+}$, $Mg^{2+}$, and $HCO_3^-$) and (2) likely anthropogenic contamination (particularly $Na^+$, $Cl^-$, and $SO_4^{2-}$). The TDS range and the major ion signature of the River Cam is similar to other anthropogenically heavily-impacted rivers (*Gaillardet et al., 1999*), exhibiting enrichment in $Na^+$ (*Figure 6d*).

Overall, ion profiles clustered substantially between the three time points, indicating characteristic temporal shifts in water chemistry. PC1 of a PCA on the solute concentrations [µmol/L] shows a strong time effect, separating spring (April) from summer (June, August) samples (*Figure 6b*). We highlighted the ten most important features (i.e. features with the largest weights) and their contributions to PC1 (*Figure 6c*).

We integrated sensor data sets on mean daily air temperature, sunshine hours and total rainfall from a public, Cambridge-based weather station (*Figure 6—figure supplement 1a–c*; *Supplementary file 1c*). Similarly, mean gauged daily Cam water discharge [$m^3s^{-1}$] of the River Cam was retrieved through publicly available records from three upstream gauging stations connected to the UK National River Flow Archive (https://nrfa.ceh.ac.uk/), together with historic measurements from 1968 onwards (*Figure 6—figure supplement 1d*).

## DNA extraction

Within 24 hr of sampling, 400 mL of refrigerated freshwater from each site was filtered through an individual 0.22 µm pore-sized nitrocellulose filter (MilliporeSigma, Burlington, MA, USA) placed on a Nalgene polysulfone bottle top filtration holder (Thermo Fisher Scientific) at −30 mbar vacuum pressure. Additionally, 400 mL de-ionised (DI) water was also filtered. We then performed DNA extractions with a modified DNeasy PowerWater protocol (Qiagen, Hilden, Germany). Briefly, filters were cut into small slices with sterile scissors and transferred to 2 mL Eppendorf tubes containing lysis beads. Homogenisation buffer PW1 was added, and the tubes subjected to ten minutes of vigorous shaking at 30 Hz in a TissueLyser II machine (Qiagen). After subsequent DNA binding and washing steps in accordance with the manufacturer's protocol, elution was done in 50 µL EB. We used Qubit dsDNA HS Assay (Thermo Fisher Scientific) to determine water DNA isolate concentrations (*Supplementary file 2a*).

## Bacterial full-length 16S rDNA sequence amplification

DNA extracts from each sampling batch and DI water control were separately amplified with V1-V9 full-length (~1.45 kbp) 16S rRNA gene primers, and respectively multiplexed with an additional

sample with a defined bacterial mixture composition of eight species (*Pseudomonas aeruginosa*, *Escherichia coli*, *Salmonella enterica*, *Lactobacillus fermentum*, *Enterococcus faecalis*, *Staphylococcus aureus*, *Listeria monocytogenes*, *Bacillus subtilis*; D6305, Zymo Research, Irvine, CA, USA) (*Figure 2*), which was previously assessed using nanopore shotgun metagenomics (*Nicholls et al., 2019*). We used common primer binding sequences 27F and 1492R, both coupled to unique 24 bp barcodes and a nanopore motor protein tether sequence (*Supplementary file 7*). Full-length 16S rDNA PCRs were performed with 30.8 µL DI water, 6.0 µL barcoded primer pair (10 µM), 5.0 µL PCR-buffer with MgCl$_2$ (10x), 5.0 µL dNTP mix (10x), 3.0 µL freshwater DNA extract, and 0.2 µL Taq (Qiagen) under the following conditions:

94˚C - 2 min
94˚C - 30 s, 60˚C - 30 s, 72˚C - 45 s (35 cycles)
72˚C - 5 min

## Nanopore library preparation

Amplicons were purified from reaction mixes with a QIAquick purification kit (Qiagen). Two rounds of alcoholic washing and two additional minutes of drying at room temperature were then performed, prior to elution in 30 µL 10 mM Tris-HCl pH 8.0 with 50 mM NaCl. After concentration measurements with Qubit dsDNA HS, twelve barcoded extracts of a given batch were pooled in equimolar ratios, to approximately 300 ng DNA total (*Supplementary file 2b*). We used KAPA Pure Beads (KAPA Biosystems, Wilmington, MA, USA) to concentrate full-length 16S rDNA products in 21 µL DI water. Multiplexed nanopore ligation sequencing libraries were then made by following the SQK-LSK109 protocol (Oxford Nanopore Technologies, Oxford, UK).

## Nanopore sequencing

R9.4.1 MinION flow cells (Oxford Nanopore Technologies) were loaded with 75 µl of ligation library. The MinION instrument was run for approximately 48 hr, until no further sequencing reads could be collected. Fast5 files were basecalled using Guppy (version 3.15) and output DNA sequence reads with Q > 7 were saved as fastq files. Various output metrics per library and barcode are summarised in *Supplementary file 2c*.

## Leptospira validation

In collaboration with Public Health England, raw water DNA isolates of the River Cam from each location and time point were subjected to the UK reference service for leptospiral testing (*Supplementary file 5*). This test is based on quantitative real-time PCR (qPCR) of 16S rDNA and *LipL32*, implemented as a TaqMan assay for the detection and differentiation of pathogenic and non-pathogenic *Leptospira* spp. from human serum. Briefly, the assay consists of a two-component PCR; the first component is a duplex assay that targets the gene encoding the outer membrane lipoprotein *LipL32*, which is reported to be strongly associated with the pathogenic phenotype. The second reaction is a triplex assay targeting a well-conserved region within the 16S rRNA gene (*rrn*) in *Leptospira* spp. Three different genomic variations correlate with pathogenic (PATH probe), intermediate (i.e. those with uncertain pathogenicity in humans; INTER probe) and non-pathogenic *Leptospira* spp. (ENVIRO probe), respectively.

## DNA sequence processing workflow

The described data processing and read classification steps were implemented using the Snakemake workflow management system (*Köster and Rahmann, 2012*) and are available on Github - together with all necessary downstream analysis scripts to reproduce the results of this manuscript (https://github.com/d-j-k/puntseq; *Urban et al., 2020*; copy archived at swh:1:rev:1408d508c807b88e0989a5252c5d904072dc3c4a).

## Read data processing

Reads were demultiplexed and adapters trimmed using Porechop (version 0.2.4, https://github.com/rrwick/porechop). The only non-default parameter set was '–check_reads' (to 50,000), to increase

the subset of reads to search for adapter sets. Next, we removed all reads shorter than 1.4 kbp and longer than 1.6 kbp with Nanofilt (version 2.5.0, https://github.com/wdecoster/nanofilt).

We assessed read statistics including quality scores and read lengths using NanoStat (version 1.1.2, https://github.com/wdecoster/nanostat), and used Pistis (https://github.com/mbhall88/pistis) to create quality control plots. This allowed us to assess GC content and Phred quality score distributions, which appeared consistent across and within our reads. Overall, we obtained 2,080,266 reads for April, 737,164 for June, and 5,491,510 for August, with a mean read quality of 10.0 (*Supplementary file 2c*).

## Benchmarking of bacterial taxonomic classifiers using nanopore reads

We used twelve different computational tools for bacterial full-length 16S rDNA sequencing read classification:

| Tool | Version | Commands |
|---|---|---|
| BLASTN (*Altschul et al., 1990*; *Camacho et al., 2009*) | v.2.9.0+ | blastn -task 'blastn' -db silva.fa -query Cam16S.fa -out Cam16S.out -outfmt '6' |
| Centrifuge (*Kim et al., 2016*) | v.1.0.4 | centrifuge -x centrifuge_silva -U Cam16S.fq -S Cam16S.out —report-file Cam16S.report |
| IDTAXA (*Murali et al., 2018*) | Implemented in R *DECIPHER* v.2.10.2 (*Wright, 2016*) | load('SILVA_SSU_r132_March2018.RData') IdTaxa(Cam16S.fa, trainingSet, strand = 'both', threshold = 0) |
| Kraken 2 (*Wood et al., 2019*; *Wood and Salzberg, 2014*) | v.2.0.7 | kraken2 —db kraken2_silva —output Cam16S.out —report Cam16S.report Cam16S.fa |
| MAPseq (*Matias Rodrigues et al., 2017*) | v.1.2.3 | mapseq Cam16S.fa silva.fa > Cam16S.out |
| MegaBLAST (*Camacho et al., 2009*; *Morgulis et al., 2008*) | v.2.9.0+ | blastn -task 'megablast' -db silva.fa -query Cam16S.fa -out Cam16S.out -outfmt '6' |
| Minimap2 (*Li, 2018*) | v.2.13-r852-dirty | minimap2 -ax map-ont -L silva.mmi Cam16S.fa > Cam16S.sam |
| Mothur (*Schloss et al., 2009*) | v.1.43.0 | align.seqs(candidate = Cam16S.fa, template = mothur.silva.nr_v132.align, processors = 1, ksize = 6, align = needleman) |
| QIIME 2 (*Bolyen et al., 2019*) | v.2019.7 | qiime feature-classifier classify-consensus-blast —i-query Cam16S.qza —i-reference-reads silva.qza —i-reference-taxonomy silva_tax.qza —o-classification Cam16S.out |
| RDP (*Wang et al., 2007*) | Implemented in R *DADA2* v.1.12.1 (*Callahan et al., 2016*) | assignTaxonomy(seqs = Cam16S.fa, refFasta = silva_nr_v132_train_set.fa.gz', tryRC = T, outputBootstraps = T, minBoot = 0) |
| SINTAX (*Edgar, 2016*) | Implemented in VSEARCH v.2.13.3 (*Rognes et al., 2016*) | vsearch -sintax Cam16S.fa -db silva.udb -tabbedout Cam16S.out -strand both -sintax_cutoff 0.5 |
| SPINGO (*Allard et al., 2015*) | v.1.3 | spingo -d silva.fa -k 8 -a -i Cam16S.fa > Cam16S.out |

## Datasets

We used nanopore sequencing data from our mock community and freshwater amplicons for benchmarking the classification tools. We therefore subsampled (a) 10,000 reads from each of the three mock community sequencing replicates, and (b) 10,000 reads from an aquatic sample (April-8; three random draws served as replicates). We then used the above twelve classification tools to classify these reads against the same database, SILVA v.132 (*Quast et al., 2013*; *Figure 2*; *Figure 2—figure supplement 1*).

### Comparison of mock community classifications

For the mock community classification benchmark, we assessed the number of unclassified reads, misclassified reads (i.e. sequences not assigned to any of the seven bacterial families), and the root mean squared error (RMSE) between observed and expected taxon abundance of the seven bacterial families. Following the detection of a strong bias towards the *Enterobacteriaceae* family across all classification tools, we also analysed RMSE values after exclusion of this family (*Figure 2b–c*).

### Comparison of river community classifications

For the aquatic sample, the number of unclassified reads were counted prior to monitoring the performance of each classification tool in comparison with a consensus classification, which we defined as majority vote across classifications from all computational workflows. We observed stable results across all three draws of 10,000 reads from the same dataset (data not shown), indicating a robust representation of the performance of each classifier.

### Memory and runtime measurements

To systematically assess the computational requirements and performance metrics of the twelve classification methods, 15 random subsamples of the same aquatic sample (April-8) were drawn. This test set involved 5 × 100, 5 × 1000 and 5 × 10,000 reads, each of which were independently classified by the different software frameworks. CPU time, average and peak memory metrics were recorded on a single computing node (*Figure 2—figure supplement 1b*). Due to their reusability, tool-specific reference index file generations were omitted from these measurements.

### Overall classification benchmark

Minimap2 performed second best at classifying the mock community (lowest RMSE), while also delivering freshwater bacterial profiles in line with the majority vote of other classification tools (*Figure 2*), in addition to providing comparably rapid speed (*Figure 2—figure supplement 1b*). To classify each of this study's full MinION data sets within a reasonable memory limit of 50 Gb, it was necessary to reduce the number of threads to 1, set the kmer size ('-k') to 15 and the minibatch size ('-K') to 25M.

## Bacterial genome analyses

### General workflow

After applying Minimap2 to the processed reads as explained above, we processed the resulting SAM files by firstly excluding all header rows starting with the '@' sign and then transforming the sets of read IDs, SILVA IDs, and alignment scores to tsv files of unique read-bacteria assignments either on the bacterial genus or family level. All reads that could not be assigned to the genus or family level were discarded, respectively. In the case of a read assignment to multiple taxa with the same alignment score, we determined the lowest taxonomic level in which these multiple taxa would be included. If this level was above the genus or family level, respectively, we discarded the read.

### Estimating the level of misclassifications and DNA contaminants

Across three independent sequencing replicates of the same linear bacterial community standard, we found that the fraction of reads assigned to unexpected genus level taxa lies at ~1% when using the Minimap2 classifier and the SILVA v.132 database.

Raw quantified DNA, PCR amplicons and sequencing read counts were considerably less abundant in DI water negative controls, as compared to actual freshwater specimens (*Supplementary file 2a*). Only the negative control of the most prolific flow cell run (August 2018) passed the relatively high confidence threshold of 37,000 sequencing reads on the family level (*Figure 3a*). Further inspection of these negative control reads revealed that their metagenomic profile closely mimicked the taxonomic classification profiles of river samples within the same sequencing batch, in addition to low-level kit contaminants like alphaproteobacteria of the *Bradyrhizobium* and *Methylobacterium* genus (*Salter et al., 2014*) which were otherwise nearly completely absent in any of the true aquatic isolates (*Supplementary file 8*).

## Determination of nanopore sequencing accuracy

Minimap2 alignments against mock community taxa were used to determine the mean read-wise nanopore sequencing accuracy for this study (92.08%), as determined by the formula: accuracy = 1 - (read mismatch length ÷ read alignment length).

These values were calculated for each of all eight species against each sequencing replicate, using the samtools (v.1.3.1) stats function (*1000 Genome Project Data Processing Subgroup et al., 2009*).

## Rarefaction and high-confidence samples

Sample-specific rarefaction curves were generated by successive subsampling of sequencing reads classified by Minimap2 against the SILVA v.132 database. For broader comparative data investigations, we chose to only retain samples that passed a conservative minimum threshold of 37,000 reads. Family and genus level species richness was hence kept at ~90% of the original values, in accordance with stable evenness profiles across a series of 100 bootstrap replicates (*Figure 3—figure supplement 1a–b*). Although we mainly present a single example rarefied dataset within this manuscript, we repeated each analysis, including PCAs, hierarchical clustering and Mantel tests, based on additional rarefied datasets to assess the stability of all results.

## Mantel test

We performed Mantel tests to compare rarefied datasets with the full dataset. We therefore compared the Euclidean distance based on Z-standardised bacterial genera between all samples with more than 37,000 reads (two-sided test with 99,999 permutations). This resulted in a Pearson correlation of 0.814 ($p = 2.1*10^{-4}$) for our main rarefied dataset (results of the Mantel test applied to the remaining three other rarefied datasets: R = 0.819 and $p = 1.0*10^{-4}$, R = 0.828 and $p = 8.0*10^{-5}$, R = 0.815 and $p = 1.4*10^{-4}$, respectively). Results of the Mantel tests applied to the genus level bacterial classifications were also similar for all four subsampled datasets (R = 0.847 and $p = 1.0*10^{-5}$, R = 0.863 and $p = 1.0*10^{-5}$, R = 0.851 and $p = 1*10^{-5}$, R = 0.856 and $p = 1.0*10^{-5}$).

## Meta-level bacterial community analyses

All classification assessment steps and summary statistics were performed in R or Python (https://github.com/d-j-k/puntseq; *Urban et al., 2020*). We used the Python package *scikit-bio* for the calculation of the Simpson index and the Shannon's diversity as well as equitability index.

## Hierarchical clustering, principal component, mixture model, and outlier analyses

Rarefied read count data was subjected to a $\log_{10}(x+1)$ transformation before hierarchical clustering using the complete linkage method. Resulting family and genus dendrograms were separated into four groups (clusters C1 - C4), while sample trees were split into two groups (separating mock communities from aquatic samples).

For PCA analyses, rarefied read count data was subjected to $\log_{10}(x+1)$ and Z-transformations. Negative control samples were removed. Mock community samples were initially removed to then be re-aligned to the eigenspace determined by the aquatic samples. We provide PCA visualisations of the four main principal components (PCs explaining > 5% variance, respectively). For each of these relevant PCs, we further highlight the ten most important features (i.e. taxa with largest weights) and their contributions to the PCs in barplots. To assess statistical differences in the PC3 component contribution between the three seasonal time points, a Kruskal-Wallis H-test with corresponding aquatic sample groupings was applied, followed by post-hoc comparisons using two-sided Mann-Whitney U rank tests.

We fit a zero-inflated log-normal mixture model of each bacterial taxon against the different time points using the fitFeatureModel function embedded in the R package *metagenomeSeq* (*Paulson et al., 2013*). As only three independent variables can be accounted for by the model (including the intercept), we chose to investigate the difference between the spring (April) and summer (June, August) months. Seven significant bacterial taxa were inspected below a nominal P-value threshold of 0.05: *Cyanobiaceae* ($1.5*10^{-5}$), *Listeriaceae* ($2.0*10^{-4}$), *Azospirillaceae* ($6.8*10^{-4}$),

*Cryomorphaceae* (1.3*10$^{-3}$), *Carnobacteriaceae* (4.3*10$^{-3}$), *Microbacteriaceae* (0.014), *Armatimonadaceae* (0.046).

To determine location and time point-specific bacterial overabundance (outlier analysis), we identified taxa which were (1.) tagged by more than 500 reads and (2.) at least five times more abundant in any single sample than in the mean of all samples combined.

## Identification of the core microbiome

The core microbiome was calculated based on rarefied read count data from four independent downsampling sets on either family or genus level (*Figure 4*; *Figure 4—figure supplement 1*) represents the most abundant taxa that showed consistent abundance profiles between samples, based on hierarchical clustering analysis on one independent rarefaction (*Figure 4a*, C2 and C4; *Figure 4— figure supplement 1a*, C3 and C2) and rarefactions with a median abundance of > 0.1%. For the genus level, only those with median abundance of > 0.2% are displayed.

## Pathogen candidate assessments

A list of 55 known bacterial pathogenic genera, spanning 37 families, was compiled for targeted sequence testing. This was done through the manual integration of curated databases and online sources, foremost using PATRIC (*Wattam et al., 2017*) and data on known waterborne pathogens (*Jin et al., 2018*; *Supplementary file 3a*). Additionally, we integrated known genera from a large wastewater reference collection (*Global Water Microbiome Consortium et al., 2019*; *Supplementary file 3b*).

To identify if DNA reads assigned to *Leptospiraceae* were more similar to sequence reads of previously identified pathogenic, intermediate or environmental *Leptospira* species, we built a neighbour-joining tree of *Leptospiraceae* reads classified in our samples data, together with sequences from reference databases (*Figure 8d*; species names and NCBI accession numbers in clockwise rotation around the tree in *Supplementary file 4d*). We matched the orientation of our reads, and then aligned them with 68 *Leptospira* reference sequences and the *Leptonema illini* reference sequence (DSM 21528 strain 3055) as an outgroup. We then built a neighbour-joining tree using Muscle v.3.8.31 (*Edgar, 2004*), excluding three reads in the 'Other Environmental' clade that had extreme branch lengths > 0.2. The reference sequences were annotated as pathogenic and saprophytic clades P1, P2, S1, S2 as recently described (*Vincent et al., 2019*). Additional published river water *Leptospira* that did not fall within these clades were included as 'Other Environmental' (*Ganoza et al., 2006*). Similarly, we constructed phylogenies for the *Legionella*, *Salmonella* and *Pseudomonas* genus, using established full-length 16S reference species sequences from NCBI (*Figure 8a–c*; *Supplementary file 4a-c*).

## Total project cost

This study was designed to enable freshwater microbiome monitoring in budget-constrained research environments. Although we had access to basic infrastructure such as pipettes, a PCR and TissueLyser II machine, as well a high-performance laptop, we wish to highlight that the total sequencing consumable costs were held below £4000 (*Supplementary file 6a*). Individual processing and sequencing costs ranged at ~£75 per sample (*Supplementary file 6b*). With the current MinION flow cell price of £720, we estimate that per-sample costs could be further reduced to as low as ~£20 when barcoding and pooling ~100 samples in the same sequencing run (*Supplementary file 6c*). Assuming near-equimolar amplicon pooling, flow cells with an output of ~5,000,000 reads can yield well over 37,000 sequences per sample and thereby surpass this conservative threshold applied here for comparative river microbiota analyses.

## Acknowledgements

We wish to thank reviewers Dr. María Mercedes Zambrano, Dr. Alejandro Sanchez-Flores, and reviewing editor Dr. Bavesh Kana for their valuable comments and improvements to this manuscript, particularly during the worldwide pandemic of COVID-19. We further thank Meltem Gürel, Christian Schwall, Jack Monahan, Eirini Vamva, Astrid Wendler, Ben Wagstaff, Elliot Brooks, Jennifer Pratscher, Rob Field, David Seilly, Mervyn Greaves, Tim Brooks, Daniel Bailey, Jenny Molloy, Michal

Filus, Aleix Lafita, Oana Stroe, Abigail Wood, Paul Saary, Jane Clarke, Fiona Gilsenan and her family, Nick Loman, Zamin Iqbal, Rob Finn, Alex Greenwood, Daniela Numberger, Julian Parkhill, Simon Frost, Sam Stubbs, Mark Holmes, Alicja Dabrowska, Alex Patto, Adrien Leger, Kim Judge, Alina Ham, Dan Fordham, Heather Martinez, Gemma Gambrill, Víctor de Lorenzo, David Sargan, Lisa Schmunk, Amanda Clare, Alejandro de Miquel Bleier and Alison Smith for helpful comments and assistance with this project. We thank Lilo and Manfred Fuchs from the Fuchs Fund for supporting Lara Urban's conference participation and presentation.

## Additional information

### Funding

| Funder | Grant reference number | Author |
|---|---|---|
| Gates Cambridge Trust | Graduate Student Fellowship | Andre Holzer<br>Maximilian R Stammnitz |
| Bill and Melinda Gates Foundation | OPP1144 | Andre Holzer<br>Maximilian R Stammnitz |
| Biotechnology and Biological Sciences Research Council | OpenPlant Fund (BBSRC BB/L014130/1) | Lara Urban<br>Andre Holzer<br>J Jotautas Baronas<br>Michael B Hall<br>Philipp Braeuninger-Weimer<br>Michael J Scherm<br>Daniel J Kunz<br>Surangi N Perera<br>Daniel E Martin-Herranz<br>Edward T Tipper<br>Susannah J Salter<br>Maximilian R Stammnitz |
| University of Cambridge | Public Engagement Starter Grant (RCUK Catalyst Seed Fund) | Lara Urban<br>Andre Holzer<br>J Jotautas Baronas<br>Michael B Hall<br>Philipp Braeuninger-Weimer<br>Michael J Scherm<br>Daniel J Kunz<br>Surangi N Perera<br>Daniel E Martin-Herranz<br>Edward T Tipper<br>Susannah J Salter<br>Maximilian R Stammnitz |
| European Bioinformatics Institute | Graduate Student Fellowship | Lara Urban<br>Michael B Hall<br>Daniel E Martin-Herranz |
| Wellcome Trust | Graduate Student Fellowship (203828/Z/16/A, 203828/Z/16/Z) | Daniel J Kunz |
| Wellcome Trust | Graduate Student Fellowship (102453/Z/13/Z) | Surangi N Perera |
| Oliver Gatty Studentship | Graduate Student Fellowship | Michael J Scherm |
| Natural Environment Research Council | Standard Grant (NE/P011659/1) | J Jotautas Baronas<br>Edward T Tipper |

The funders had no role in study design, data collection and interpretation, or the decision to submit the work for publication.

### Author contributions

Lara Urban, Andre Holzer, Data curation, Software, Formal analysis, Validation, Investigation, Visualization, Methodology, Writing - original draft, Project administration, Writing - review and editing; J Jotautas Baronas, Resources, Data curation, Formal analysis, Investigation, Methodology, Writing -

review and editing; Michael B Hall, Software, Formal analysis, Investigation, Methodology, Writing - review and editing; Philipp Braeuninger-Weimer, Conceptualization, Data curation, Software, Formal analysis, Funding acquisition, Investigation, Methodology, Writing - review and editing; Michael J Scherm, Resources, Data curation, Formal analysis, Validation, Investigation, Methodology, Writing - review and editing; Daniel J Kunz, Data curation, Software, Formal analysis, Validation, Investigation, Methodology, Writing - review and editing; Surangi N Perera, Conceptualization, Resources, Data curation, Validation, Investigation, Methodology, Writing - review and editing; Daniel E Martin-Herranz, Conceptualization, Data curation, Software, Formal analysis, Funding acquisition, Validation, Investigation, Methodology, Writing - review and editing; Edward T Tipper, Resources, Supervision, Validation, Methodology, Writing - review and editing; Susannah J Salter, Formal analysis, Supervision, Validation, Investigation, Visualization, Methodology, Project administration, Writing - review and editing; Maximilian R Stammnitz, Conceptualization, Resources, Data curation, Software, Formal analysis, Supervision, Funding acquisition, Validation, Investigation, Visualization, Methodology, Writing - original draft, Project administration, Writing - review and editing

**Author ORCIDs**
Lara Urban (ID) https://orcid.org/0000-0002-5445-9314
Andre Holzer (ID) http://orcid.org/0000-0003-2439-6364
J Jotautas Baronas (ID) https://orcid.org/0000-0002-4027-3965
Michael B Hall (ID) https://orcid.org/0000-0003-3683-6208
Philipp Braeuninger-Weimer (ID) https://orcid.org/0000-0001-8677-1647
Michael J Scherm (ID) http://orcid.org/0000-0002-3289-9159
Daniel J Kunz (ID) https://orcid.org/0000-0003-3597-6591
Surangi N Perera (ID) https://orcid.org/0000-0003-4827-9242
Daniel E Martin-Herranz (ID) https://orcid.org/0000-0002-2285-3317
Edward T Tipper (ID) https://orcid.org/0000-0003-3540-3558
Susannah J Salter (ID) https://orcid.org/0000-0003-3898-8504
Maximilian R Stammnitz (ID) https://orcid.org/0000-0002-1704-9199

**Decision letter and Author response**
Decision letter https://doi.org/10.7554/eLife.61504.sa1
Author response https://doi.org/10.7554/eLife.61504.sa2

## Additional files

**Supplementary files**
• Supplementary file 1. Summary of samples and metadata. (a) Sampling locations. (b) Environmental metadata by sample. (c) Environmental metadata by time point.

• Supplementary file 2. Summary of raw DNA, amplicon and sequencing yields. (a) Water DNA extraction yields. (b) Full-length 16S PCR amplicon extraction yields. (c) Nanopore sequencing read metrics.

• Supplementary file 3. Summary of pathogen and wastewater bacterial genera tested. (a-b) List of pathogen (a) and wastewater (b) candidate bacterial genera.

• Supplementary file 4. Summary of reference sequences for high-resolution pathogen mapping. (a-d) References and NCBI accessions for *Legionella* (a), *Salmonella* (b), *Pseudomonas* (c), and *Leptospira* (d).

• Supplementary file 5. Summary of multi-species *Leptospira* quantifications by Taqman qPCR.

• Supplementary file 6. Summary of project costs. (a) Basic sequencing workflow cost estimate. (b) Cost estimate per sample, based on a 12-plex MinION sequencing run. (c) Projected cost estimate per sample, based on a 100-plex MinION sequencing run.

• Supplementary file 7. Summary of full-length 16S primer sequences (5' - 3').

- Supplementary file 8. Summary of negative controls. (a-c) Relative classification output per sample (%), sorted by negative control abundances in April (a), June (b), and August (c).

- Transparent reporting form

## Data availability

Sequencing datasets generated and analysed during this study are available from the European Nucleotide Archive (ENA), under project accession PRJEB34900. Hydrochemical measurements are available in Supplementary file 1 of this article, as well as weather and river flow speed summary data from two public repositories, https://www.cl.cam.ac.uk/research/dtg/weather/ and https://nrfa. ceh.ac.uk/. Leptospira TaqMan qPCR diagnostics results, as obtained by Public Health England, are available in Supplementary file 5. There are no restrictions on data availability.

The following dataset was generated:

| Author(s) | Year | Dataset title | Dataset URL | Database and Identifier |
|---|---|---|---|---|
| Urban L, Holzer A, Stammnitz M.R | 2020 | Freshwater monitoring by nanopore sequencing | https://www.ebi.ac.uk/ ena/data/view/ PRJEB34900 | EBI European Nucleotide Archive, PRJEB34900 |

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
