## [Decision Letter]

**Acceptance summary:**

In this work the authors analyze freshwater bacterial communities using nanopore DNA sequencing and optimized experimental and bioinformatics guidelines. This workflow can efficiently assess river microbiomes and provides a cost-effective and accessible strategy for environmental monitoring of microbial communities.

**Decision letter after peer review:**

Thank you for submitting your article "Freshwater monitoring by nanopore sequencing" for consideration by *eLife*. Your article has been reviewed by two peer reviewers, including María Mercedes Zambrano as the Reviewing Editor and Reviewer #1, and the evaluation has been overseen by Bavesh Kana as the Senior Editor. The following individual involved in review of your submission has agreed to reveal their identity: Alejandro Sanchez-Flores (Reviewer #2).

The reviewers have discussed the reviews with one another and the Reviewing Editor has drafted this decision to help you prepare a revised submission.

Summary:

The authors present a survey of the bacterial community in the Cam River in Cambridge, UK, using Nanopore DNA sequencing, one of the latest DNA sequencing technologies. They profile microbial communities along the river, correlate with physiochemical parameters and identify potential pathogens and sewage signals. The work provides standardized protocols and bioinformatics tools for analysis of bacteria in freshwater samples, with the aim of providing a low-cost and optimized workflow that can be applied for the monitoring of complex aquatic microbiomes.

Essential revisions:

The work is well-carried out and timely, but the document could improve by addressing the following:.

1) Please comment on why many of the June samples failed to provide sufficient sequence information, especially since not all of them had low yields (Supplementary file 2 and Figure 2—figure supplement 2).

2) It would be helpful if the authors could mention the amount (or proportion) of their sequenced 16S amplicons that provided species-level identification, since this is one of the advantages of nanopore sequencing.

3) It is not entirely clear how the authors define their core microbiome. Are they reporting mainly the most abundant taxa (dominant core microbiome), and would this change if you look at a taxonomic rank below the family level? How does the core compare, for example, with other studies of this same river?

4) Please consider revising the amount of information in some of the figures (such as Figure 2 and Figure 3). The resulting images are tiny, the legends become lengthy and the overall impact is reduced. Consider splitting these or moving some information to the supplements.

5) Given that the authors claim to provide a simple, fast and optimized workflow it would be good to mention how this workflow differs or provides faster and better analysis than previous work using amplicon sequencing with a MinION sequencer.

They also mention that nanopore sequencing is an "inexpensive, easily adaptable and scalable framework" The term "inexpensive" doesn't seem appropriate since it is relative. In addition, they should also discuss that although it is technically convenient in some aspects compared to other sequencers, there are still protocol steps that need certain reagents and equipment that is similar or the same to those needed for other sequencing platforms. Common bottlenecks such as DNA extraction methods, sample preservation and the presence of inhibitory compounds should be mentioned.

---

## [Author Response]

Essential revisions:The work is well-carried out and timely, but the document could improve by addressing the following:.1) Please comment on why many of the June samples failed to provide sufficient sequence information, especially since not all of them had low yields (Supplementary file 2 and Figure 2—figure supplement 2).

An extended paragraph about experimental intricacies of our study has been added to the Discussion. It has also been slightly restructured to give a better and wider overview of how future freshwater monitoring studies using nanopore sequencing can be improved.

We wish to highlight that all three MinION sequencing runs here analysed feature substantially higher data throughput than that of any other recent environmental 16S rRNA sequencing study with nanopore technology, as recently reviewed by Latorre-Pérez et al. (Biology Methods and Protocols 2020, doi:10.1093/biomethods/bpaa016). One of this work's sequencing runs has resulted in lower read numbers for water samples collected in June 2018 (~0.7 Million), in comparison to the ones collected in April and August 2018 (~2.1 and ~5.5 Million, respectively). While log-scale variabilities between MinION flow cell throughput have been widely reported for both 16S and shotgun metagenomics approaches (e.g. see Latorre-Pérez et al.), the count of barcode-specific 16S reads is nevertheless expected to be correlated with the barcode-specific amount of input DNA within a given sequencing run. As displayed in Figure 2—figure supplement 2B, we see a positive, possibly logarithmic trend between the DNA concentration after 16S rDNA amplification and number of reads obtained.

With few exceptions (April-6, April-9.1 and Apri-9.2), we find that sample pooling with original 16S rDNA concentrations of <inline-graphic mime-subtype="png" mimetype="image" xlink:href="media/image1.png" />4 ng/µl also results in the surpassing of the here-set (conservative) minimum read threshold of 37,000 for further analyses. Conversely, all June samples that failed to reach 37,000 reads did not pass the input concentration of 4 ng/µl, despite our attempt to balance their quantity during multiplexing.

We reason that such skews in the final barcode-specific read distribution would mainly arise from small concentration measurement errors, which undergo subsequent amplification during the upscaling with comparably large sample volume pipetting. While this can be compensated for by high overall flow cell throughput (e.g. see August-2, August-9.1, August-9.2), we think that future studies with much higher barcode numbers can circumvent this challenge by leveraging an exciting software solution: real-time selective sequencing via “Read Until”, as developed by Loose, Malla and Stout, 2016. In the envisaged framework, incoming 16S read signals would be in situ screened for the sample-barcode which in our workflow is PCR-added to both the 5' and 3' end of each amplicon. Overrepresented barcodes would then be counterbalanced by targeted voltage inversion and pore "rejection" of such reads, until an even balance is reached. Lately, such methods have been computationally optimised, both through the usage of GPUs (Payne et al., 2020) and raw electrical signals (Kovaka et al., 2020).

2) It would be helpful if the authors could mention the amount (or proportion) of their sequenced 16S amplicons that provided species-level identification, since this is one of the advantages of nanopore sequencing.

We wish to emphasize that we intentionally refrained from reporting the proportion of 16S rRNA reads that could be classified at species level, since we are wary of any automated species level assignments even if the full-length 16S rRNA gene is being sequenced. While we list the reasons for this below, we appreciate the interest in the theoretical proportion of reads at species level assignment. We therefore re-analyzed our dataset, and now also provide the ratio of reads that could be classified at species level using Minimap2 (Discussion).

To this end, we classified reads at species level if the species entry of the respective SILVA v.132 taxonomic ID was either not empty, or neither *uncultured bacterium* nor *metagenome*. Therefore, many unspecified classifications such as *uncultured species* of some bacterial genus are counted as species-level classifications, rendering our approach lenient towards a higher ratio of species level classifications. Still, the species level classification ratios remain low, on average at 16.2 % across all included river samples (genus-level: 65.6 %, family level: 76.6 %). The mock community, on the other hand, had a much higher species classification rate (> 80 % in all three replicates), which is expected for a well-defined, well-referenced and divergent composition of only eight bacterial taxa, and thus re-validates our overall classification workflow.

On a theoretical level, we mainly refrain from automated across-the-board species level assignments because:

1) Many species might differ by very few nucleotide differences within the 16S amplicon; distinguishing these from nanopore sequencing errors (here ~8 %) remains challenging;

2) Reference databases are incomplete and biased with respect to species level resolution, especially regarding certain environmental contexts; it is likely that species assignments would be guided by references available from more thoroughly studied niches than freshwater.

Other recent studies have also shown that across-the-board species-level classification is not yet feasible with 16S nanopore sequencing, for example in comparison with Illumina data (Acharya et al., 2019) which showed that “more reliable information can be obtained at genus and family level”, or in comparison with longer 16S-ITS-23S amplicons (Cusco et al., 2018), which “remarkably improved the taxonomy assignment at the species level”.

3) It is not entirely clear how the authors define their core microbiome. Are they reporting mainly the most abundant taxa (dominant core microbiome), and would this change if you look at a taxonomic rank below the family level? How does the core compare, for example, with other studies of this same river?

The here-presented core microbiome indeed represents the most abundant taxa, with relatively consistent profiles between samples. We used hierarchical clustering (Figure 4A, C2 and C4) on the bacterial family level, together with relative abundance to identify candidate taxa. Filtering these for median abundance > 0.1% across all samples resulted in 27 core microbiome families. To clarify this for the reader, we have added a new paragraph to the Materials and methods section.

We have also performed the same analysis on the bacterial genus level and now display the top 27 most abundant genera (median abundance > 0.2%), together with their corresponding families and hierarchical clustering analysis in the new Figure 4—figure supplement 1. Overall, high robustness is observed with respect to the families of the core microbiome: out of the top 16 core families (Figure 4B), only the NS11-12 marine group family is not represented by the top 27 most abundant genera (Figure 4—figure supplement 1B). We reason that this is likely because its corresponding genera are composed of relatively poorly resolved references of uncultured bacteria, which could thus not be further classified.

To the best of our knowledge, there are only two other reports that feature metagenomic data of the River Cam and its wastewater influx sources (Rowe et al., 2016; Rowe et al., 2017). While both of these primarily focus on the diversity and abundance of antimicrobial resistance genes using Illumina shotgun sequencing, they only provide limited taxonomic resolution on the river's core microbiome. Nonetheless, Rowe et al., 2016, specifically highlighted *Sphingobium* as the most abundant genus in a source location of the river (Ashwell, Hertfordshire). This genus belongs to the family of *Sphingomonadaceae*, which is also among the five most dominant families identified in our dataset. It thus forms part of what we define as the core microbiome of the River Cam (Figure 4B), and we have therefore highlighted this consistency in our manuscript's Discussion.

4) Please consider revising the amount of information in some of the figures (such as Figure 2 and Figure 3). The resulting images are tiny, the legends become lengthy and the overall impact is reduced. Consider splitting these or moving some information to the supplements.

To follow this advice, we have split Figure 2 into two less compact figures. We have moved more detailed analyses of our classification tool benchmark to the supplement (now Figure 1—figure supplement 1). Figure 1—figure supplement 1 notably also contains a new summary of the systematic computational performance measurements of each classification tool.

Moreover, we here suggest that the original Figure 3 may be divided into two figures: one to visualise the sequencing output, data downsampling and distribution of the most abundant families (now Figure 3), and the other featuring the clustering of bacterial families and associated core microbiome (now Figure 4). We think that both the data summary and clustering/core microbiome analyses are of particular interest to the reader, and that they should be kept as part of the main analyses rather than the supplement – however, we are certainly happy to discuss alternative ideas with the reviewers and editors.

5) Given that the authors claim to provide a simple, fast and optimized workflow it would be good to mention how this workflow differs or provides faster and better analysis than previous work using amplicon sequencing with a MinION sequencer.

Data throughput, sequencing error rates and flow cell stability have seen rapid improvements since the commercial release of MinION in 2015. In consequence, bioinformatics community standards regarding raw data processing and integration steps are still lacking, as illustrated by a thorough recent benchmark of fast5 to fastq format "basecalling" methods (Wick, Judd and Holt, 2019).

Early on during our analyses, we noticed that a plethora of bespoke pipelines have been reported in recent 16S environmental surveys using MinION (e.g. Kerkhof et al., 2017; Cusco et al., 2018; Acharya et al., 2019; Nygaard et al., 2020). This underlines a need for more unified bioinformatics standards of (full-length) 16S amplicon data treatment, while similar benchmarks exist for short-read 16S metagenomics approaches, as well as for nanopore shotgun sequencing (e.g. Ye et al., Cell 2019, doi:10.1016/j.cell.2019.07.010; Latorre-Pérez et al., 2020).

By adding a thorough speed and memory usage summary (new Figure 1—figure supplement 1B), in addition to our (mis)classification performance tests based on both mock and complex microbial community analyses, we provide the reader with a broad overview of existing options. While the widely used Kraken 2 and Centrifuge methods provide exceptional speed, we find that this comes with a noticeable tradeoff in taxonomic assignment accuracy. We reason that Minimap2 alignments provide a solid compromise between speed and classification performance, with the MAPseq software offering a viable alternative should memory usage limitation apply to users.

We intend to extend this benchmarking process to future tools, and to update it on our GitHub page (https://github.com/d-j-k/puntseq). This page notably also hosts a range of easy-to-use scripts for employing downstream 16S analysis and visualization approaches, including ordination, clustering and alignment tests.

The revised Discussion now emphasises the specific advancements of our study with respect to freshwater analysis and more general standardisation of nanopore 16S sequencing, also in contrast to previous amplicon nanopore sequencing approaches in which only one or two bioinformatics workflows were tested.

They also mention that nanopore sequencing is an "inexpensive, easily adaptable and scalable framework" The term "inexpensive" doesn't seem appropriate since it is relative. In addition, they should also discuss that although it is technically convenient in some aspects compared to other sequencers, there are still protocol steps that need certain reagents and equipment that is similar or the same to those needed for other sequencing platforms. Common bottlenecks such as DNA extraction methods, sample preservation and the presence of inhibitory compounds should be mentioned.

We agree with the reviewers that “inexpensive” is indeed a relative term, which needs further clarification. We therefore now state that this approach is “cost-effective” and discuss future developments such as the 96-sample barcoding kits and Flongle flow cells for small-scale water diagnostics applications, which will arguably render lower per-sample analysis costs in the future (Discussion).

Other investigators (e.g. Boykin et al., 2019; Acharya et al., Water Technology 2020, doi:10.1016/j.watres.2020.116112) have recently shown that the full application of DNA extraction and in-field nanopore sequencing can be achieved at comparably low expense: Boykin et al. studied cassava plant pathogens using barcoded nanopore shotgun sequencing, and estimated costs of ~45 USD per sample, while we calculate ~100 USD per sample in this study. Acharya et al. undertook in situ water monitoring between Birtley, UK and Addis Ababa, Ethiopia, estimated ~75-150 USD per sample and purchased all necessary equipment for ~10,000 GBP – again, we think that this lies roughly within a similar range as our (local) study's total cost of ~3,670 GBP (Supplementary file 6).

The revised manuscript now mentions the possibility of increasing sequencing yield by improving DNA extraction methods, by taking sample storage and potential inhibitory compounds into account in the planning phase (Discussion).